# UniMind: Unleashing the Power of LLMs for Unified Multi-Task Brain Decoding

## Abstract

Decoding human brain activity from electroencephalography (EEG) signals is a central challenge at the intersection of neuroscience and artificial intelligence, enabling diverse applications in mental state assessment, clinical monitoring, and human–machine interaction. Recent efforts have extensively investigated building EEG-based pretrained encoders for generalized brain decoding through large-scale training on multiple datasets. However, most of these approaches still struggle to achieve satisfactory performance without **task-specific tuning**, owing to the pronounced inherent heterogeneity across decoding tasks. To address these challenges, we present *UniMind*, a general-purpose EEG foundation model for unified multi-task brain decoding by uniquely unleashing the power of LLMs to comprehend complex neural patterns. UniMind enjoys several merits. First, we design a **Neuro-Language Connector** to transform the spatiotemporal neural patterns of EEG data into LLM-understandable representations. Second, a **Task-aware Query Selection** module is proposed to inject task-awareness into the cross-modal understanding by dynamically generating task-adaptive query tokens, enabling the learning of task-relevant neural patterns across diverse tasks. Extensive experiments across 10 datasets demonstrate that UniMind substantially outperforms state-of-the-art multi-task decoding models (**11%** gain on average), while also offering valuable neuroscientific insights into neural functional correlations across tasks. The code will be made publicly available.

## 1 Introduction

Electroencephalography (EEG) is a widely adopted technique used to measure and record electrical activity in the brain, where electrodes are placed on the scalp to detect and amplify the brain's electrical signals. EEG plays a crucial role in Brain-Computer Interfaces (BCI) by providing a non-invasive, real-time measure of brain activity, and also serves as a powerful tool for investigating brain's perceptual mechanisms. By analyzing neural patterns in EEG signals, many studies have been conducted to decode specific brain states, demonstrating strong potential across diverse applications such as seizure detection (Boonyakitanont et al., 2020), sleep stage classification (Aboalayon et al., 2016; Tang et al., 2025), motor imagery recognition (Amin et al., 2019), abnormality detection (Roy et al., 2019), emotion analysis (Suhaimi et al., 2020; Rehman et al., 2025), and acute stress detection (Sharma et al., 2022).

Despite the advances in EEG signal decoding, numerous deep learning models (Jing et al., 2023; Dar et al., 2020; Yang et al., 2022; 2023b; Peh et al., 2022; Song et al., 2021) have been confined to task-specific paradigms, due to the variations in EEG signal formats across different tasks. While effective for intended tasks, these models struggle to generalize to new tasks. Recently, pre-trained EEG encoders such as BIOT (Yang et al., 2023a) and LaBraM (Jiang et al., 2024) have attempted to learn robust representations of EEG signals through large-scale self-supervised pre-training with massive unlabeled EEG data. While

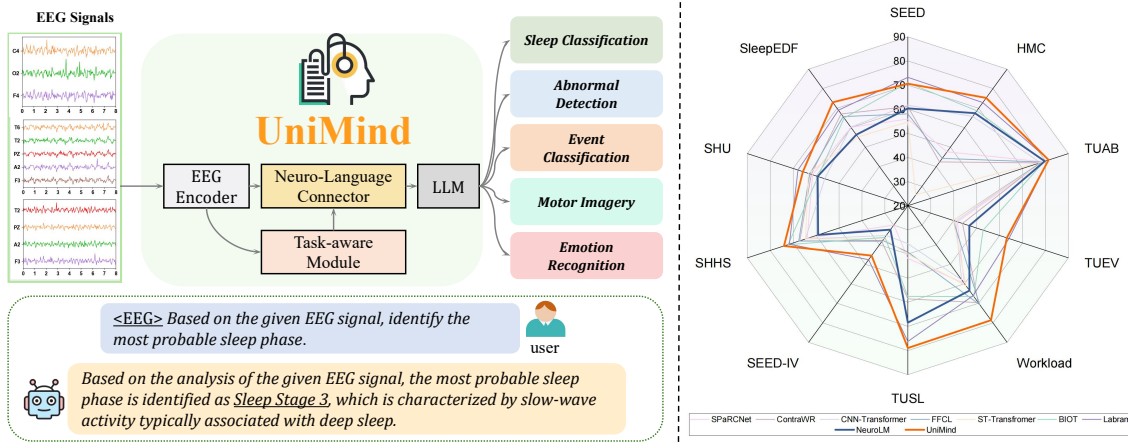

Figure 1: UniMind leverages LLMs to interpret brain signals, enabling multi-task EEG decoding without fine-tuning and demonstrating strong adaptability to task variability.

compatible with various EEG signal formats, these encoders require additional full fine-tuning with task-specific heads for downstream tasks, causing large-scale tuning effort and extra computational overhead.

The above limitations underscore the urgent need for general-purpose EEG foundation models that can unifiedly handle diverse decoding applications without the need of task-specific fine-tuning. Inspired by the success of Large Language Models (LLMs) in interpreting diverse modalities like vision and audio (Dai et al., 2023; Lyu et al., 2023; Han et al., 2024), we propose to harness the powerful reasoning capabilities of LLMs to analyze brain signal time-series data. Recently proposed NeuroLM (Jiang et al., 2025) have made an initial attempt by conducting adversarial domain-level alignment to integrate EEG and LLMs, which demonstrates the possibilities of LLMs in EEG understanding. However, its performance has not yet fully matched that of single-task methods, showing a performance deficit of more than 10% on datasets such as SEED (Zheng & Lu, 2015), TUEV (Harati et al., 2015), and SleepEDF (Aboalayon et al., 2016). To boost LLMs' potential for decoding neural patterns and unifying diverse EEG tasks, two critical challenges need to be addressed:

**(1) Huge modality gap between neural signals and LLM.** Unlike the rich, structured features of text, EEG signals are characterized by their high noise, sparse information, and complex neural patterns that are difficult for LLMs to directly perceive. An essential prerequisite for LLMs to comprehend and reason with EEG data is effective alignment with LLM's input space. Previous methods have attempted this through adversarial training at the domain level, but the modality gap remains unresolved due to neglected fine-grained cross-modal relationships. *The core issue remains how to conduct effective **cross-modality bridging** to alleviate the fundamental gap between EEG and LLMs.*

**(2) Heterogeneity across EEG decoding tasks.** EEG tasks show significant heterogeneity (Kaplan et al., 2005) due to varied configurations and cognitive mechanisms, resulting in diverse signal characteristics across datasets, such as electrode channels, trial durations, and spatiotemporal brain activity. Given this, purely task-agnostic mixed-task tuning may lead to degraded decoding performance on specific tasks. Therefore, a task-aware mechanism is urgently required for generalizable brain decoding in the multi-task setting. Moreover, task-agnostic methods overlook neural correlations across tasks, limiting insights into brain cognition. Therefore, *how to develop a **task-aware mechanism** for achieving unified multi-task learning robust to EEG task heterogeneity without sacrificing individual task performance,* remains an open challenge.

To address these challenges, we introduce **UniMind**, a general-purpose brain foundation model for unified multi-task brain decoding by unlocking the potential of LLMs in understanding brain patterns, as shown in Figure 1. To the best of our knowledge, it is the first multi-task EEG decoding model to match the

performance of single-task approaches in one unified model. To narrow the modality gap, we propose a **Neuro-Language Connector** (NLC) which acts as a compact, trainable bridge between the EEG encoder and the LLM, condensing essential brain patterns from sparse EEG data in a semantically meaningful way for the LLM to interpret. Specifically, to leverage the spatio-temporal nature of EEG signals (Liu et al., 2022; Wang et al., 2022), the NLC adopts a dual-branch architecture, with learnable query tokens that separately aggregate temporal dynamics and spatial dependencies from neural signals via cross-attention. The aggregated features are then aligned and mapped to the semantic space of a frozen LLM. By doing this, NLC transforms spatio-temporal neural patterns into interpretable features for LLMs, thereby enabling seamless neuro-language integration. To facilitate effective multi-task learning across heterogeneous EEG tasks, we propose a **Task-aware Query Selection Module** (TQS) to generate task-adaptive query tokens. TQS maintains spatial and temporal query pools containing multiple query tokens. A router mechanism is used to dynamically look up task-relevant queries from query pools based on input features. By allowing each task to adaptively choose its own queries, TQS promotes knowledge sharing and mutual enhancement across related tasks while mitigating interference from conflicting ones. Moreover, through the task selection mechanism, we uncover how diverse tasks are functionally organized across the brain by examining inter-task correlations, shedding light on how the brain regulates different cognitive functions. Our contribution can be summarized as follows:

- We propose **UniMind**, a general-purpose brain foundation model for unified multi-task brain decoding by integrating a spatio-temporal cross-modality bridging strategy between EEG and language, along with a task-aware mechanism.

- We propose a dual-branch neuro-language connector that encodes the temporal and spatial patterns of EEG signals into LLM-interpretable representations, along with a task-aware module that generates adaptive queries for task-relevant representations.

- Experiments demonstrate that UniMind outperforms the existing best multi-task decoding model by **11%** on average and is the first to achieve comparable or even superior performance to single-task decoding models across various tasks.

- Insights from a neuroscientific perspective are provided via query visualizations, revealing shared neural mechanisms and offering empirical support for knowledge sharing in multi-task brain decoding process.

## 2 METHOD

As illustrated in Figure 2, UniMind empowers the LLM to understand brain signals from heterogeneous tasks by integrating two modules. (1) The **Neuro-Language Connector** (NLC) (Section 2.1) aims to bridge the modality gap between neural signals and language models by learning query tokens to interpret brain patterns. (2) The **Task-aware Query Selection Module** (TQS) (Section 2.2) aims to enhance task adaptability by dynamically learning task-adaptive query tokens for feature selection across heterogeneous EEG tasks.

### 2.1 NEURO-LANGUAGE CONNECTOR

EEG signals are characterized by temporal dynamics and multichannel structures (Wang et al., 2022; Song et al., 2021). To leverage these neural properties, the neuro-language connector is designed to condense and interpret spatiotemporal neural characteristics from noisy EEG signals. Next, we present the EEG encoder for extracting neural representations, followed by the design of our neuro-language connector.

**EEG Encoder.** Given an EEG signal sample $X \in \mathbb{R}^{C \times S}$, where $C$ and $S$ respectively denote the number of electrode channels and temporal sampling points, we adopt a pre-trained EEG encoder from LaBraM (Jiang et al., 2024) to transform EEG signals with varying channels and lengths into standard token sequences.

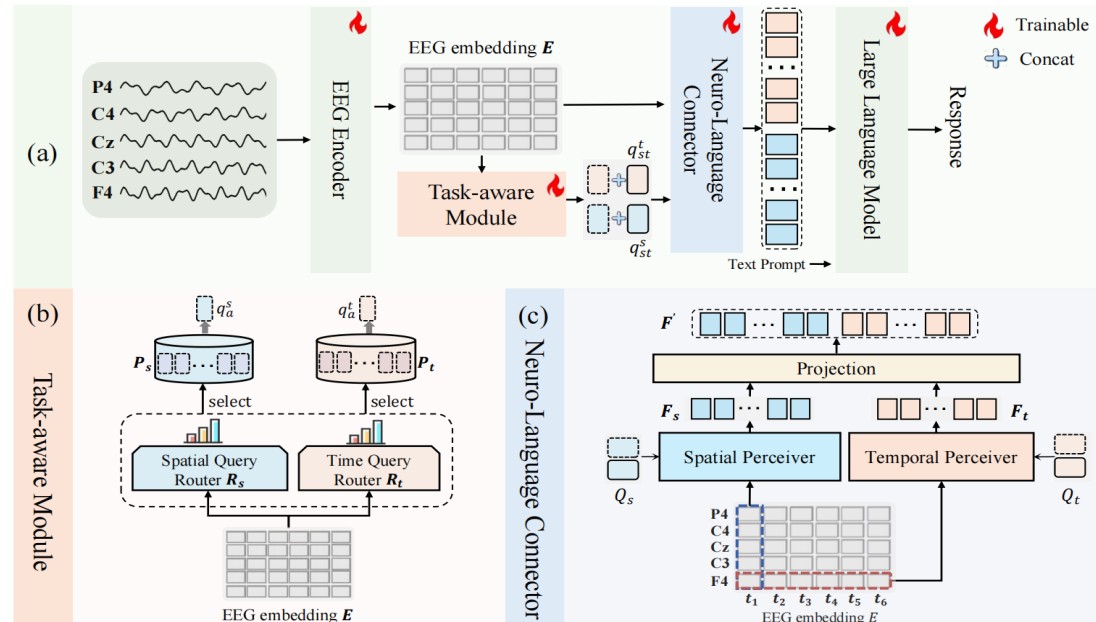

Figure 2: Overview of the UniMind architecture. EEG signals are encoded into EEG embeddings $\boldsymbol{E}$, which are processed by the TQS to extract task-adaptive queries. These queries are combined with static queries and jointly processed with $\boldsymbol{E}$ by the NLC, which aligns spatio-temporal neural features with the LLM's semantic space. The resulting embeddings, together with task prompts, guide the LLM to generate text output.

Specifically, each EEG channel is divided into non-overlapping patches using a sliding window of length $t$, resulting in $T = \lfloor \frac{S}{t} \rfloor$ patches per channel. Finally, the signal is encoded as a sequence of patch token embeddings, with a prepended class token, resulting in $\boldsymbol{E} \in \mathbb{R}^{(C \times T + 1) \times D_e}$.

**Neuro-Language Connector.** To bridge the gap between the LLM and EEG signals with sparsity and low signal-to-noise ratio, the NLC aggregates spatio-temporal features in a decoupled way via cross-attention. Specifically, given the EEG embeddings $\boldsymbol{E}$ from the EEG encoder, the NLC designs two sets of learnable queries: temporal queries $Q_t \in \mathbb{R}^{n_{qt} \times D_e}$ and spatial queries $Q_s \in \mathbb{R}^{n_{qs} \times D_e}$, where $n_{qt}$ and $n_{qs}$ denote the number of temporal queries and spatial queries, respectively. $Q_t$ captures temporal neural dynamics by attending to time-varying patterns within each channel, while $Q_s$ explores spatial dependencies by analyzing spatial-wise channel activation patterns at each time step. We employ a cross-attention mechanism to discover and aggregate key brain patterns from noisy EEG signals, with the EEG token sequence serving as Keys and Values and learnable query tokens are Queries. Formally:

$$\boldsymbol{F}_t = \text{CrossAttn}(Q_t, \boldsymbol{E}_t) \in \mathbb{R}^{n_{qt} \times C \times D_e}, \quad \boldsymbol{F}_s = \text{CrossAttn}(Q_s, \boldsymbol{E}_s) \in \mathbb{R}^{n_{qs} \times T \times D_e}, \quad (1)$$

where $\boldsymbol{E}_t \in \mathbb{R}^{C \times T \times D_e}$ and $\boldsymbol{E}_s \in \mathbb{R}^{T \times C \times D_e}$ are the reshaped EEG embeddings for temporal and spatial attention, respectively. $\boldsymbol{F}_t$ and $\boldsymbol{F}_s$ are the temporally and spatially condensed EEG embeddings after attention-based interaction, which are then concatenated with the initial class token $\boldsymbol{F}_{\text{cls}} \in \mathbb{R}^{1 \times D_e}$, resulting in the neuro-semantic embeddings $\boldsymbol{F} = \text{Concat}(\boldsymbol{F}_t, \boldsymbol{F}_s, \boldsymbol{F}_{\text{cls}}) \in \mathbb{R}^{(n_{qt} \times C + n_{qs} \times T + 1) \times D_e}$.

Finally, the neuro-semantic embeddings are projected to the embedding space of LLM to make them linguistically understandable to the LLM, resulting in $\boldsymbol{F}' \in \mathbb{R}^{(n_{qt} \times C + n_{qs} \times T + 1) \times D_L}$, where $D_L$ denotes the LLM hidden size. By transforming spatiotemporal neural patterns into semantically structured representations, the NLC help LLMs interpret neural signals based to their underlying neurophysiological characteristics.

## 2.2 TASK-AWARE QUERY SELECTION MODULE

Integrating multiple EEG tasks into one model is challenging due to heterogeneous signal characteristics across tasks. The shared architecture struggles to manage task heterogeneity, leading to degraded performance on specific tasks. Therefore, we introduce the TQS that integrates a task-aware mechanism to the cross-modality bridging process, dynamically learning query tokens for task-adaptive neural pattern decoding.

To achieve dynamic query learning, the TQS introduces a query pool $P$ that contains a series of learnable query tokens $P = [P_1, P_2, \cdots, P_{N_q}]$, and a query router $R$ for query selection. Specifically, TQS contains two branches for respective temporal and spatial query learning, denoted as $P_t, R_t$ and $P_s, R_s$. The query router $R$ generates routing scores $S$ over the query tokens in $P$ conditioned on the input EEG embedding $E$. The routing scores $S$ indicate how well each query fits to the input EEG features, allowing the model to generate task-adaptive queries tailored for the sample from the current task. Subsequently, the task-adaptive queries $q_a$ are selected from $P$ based on routing scores:

$$q_a^i = \text{TopK}(P_i, S = R_i(E)), \quad i \in \{t, s\}, \quad q_a^i \in \mathbb{R}^{n_i \times D_e}, \tag{2}$$

where $n_t$ and $n_s$ denote the number of task-adaptive queries for the temporal ($t$) and spatial ($s$) modalities, respectively. Notably, we incorporate an additional static query token $q_{st} \in \mathbb{R}^{1 \times D_e}$, which captures the stable characteristics of each task. The final query token sequence can be formulated as: $Q_i = [q_a^i; q_{st}^i] \in \mathbb{R}^{(n_i+1) \times D_e}, i \in \{t, s\}$. The resulting queries $Q_t$ and $Q_s$ are then fed into the NLC to extract task-relevant temporal and spatial features from EEG signals. TQS enables each task to select its own queries adaptively, fostering knowledge sharing across related tasks while reducing the interference risk from conflicting ones.

## 2.3 MULTI-TASK DECODING WITH LLMs

**Multi-task instruction tuning dataset.** We adopt instruction tuning for training the multi-task foundation model, which involves collecting multiple brain decoding tasks and creating a unified instruction dataset that encompasses them all. To this end, we construct an instruction tuning dataset specifically tailored for brain decoding, comprising five diverse task types and containing a total of 929k samples. To further enrich the instruction diversity, we design ten distinct prompts for each task type. For every sample, one of these prompts is randomly selected and paired as a task-specific instruction. Further details regarding the instruction templates are provided in the **Appendices**.

**LLM Training.** For training, we unfreeze the EEG encoder, TQS, and NLC to fully adapt them to the downstream tasks. Meanwhile, the LLM is fine-tuned using LoRA, allowing parameter-efficient adaptation while keeping the majority of the LLM parameters frozen. We cast EEG decoding as conditional language modeling. Given EEG-derived neural-semantic features $F'$, a separator token [SEP], and task instruction $P$, the input is $X = [F', [\text{SEP}], P]$. The model predicts the target sequence $Y = \{t_a\}$ and is trained with causal cross-entropy loss $\mathcal{L} = -\sum_{i=1}^{L} \log p(t_{a,i} \mid X, t_{a,<i})$.

## 3 EXPERIMENTS

### 3.1 EXPERIMENTAL SETUP

**Datasets & Metrics.** To comprehensively evaluate *UniMind*, we adopt ten publicly available EEG datasets across five task domains: sleep stage classification (HMC (Alvarez-Estevez & Rijsman, 2021), SleepEDF (Kemp et al., 2000), SHHS (Quan et al., 1997)), emotion recognition (SEED (Zheng & Lu, 2015), SEED-IV (Zheng et al., 2019)), clinical anomaly detection (TUAB (Harati et al., 2015), TUEV (Harati et al., 2015), TUSL (von Weltin et al., 2017)), cognitive workload classification (Workload (Zyma et al., 2019)), and motor imagery classification (SHU (Ma et al., 2022)). Following previous works, we consider both

Table 1: Performance comparison of multi-task models in terms of Balanced Accuracy.

| Model | Multi-subject | | | Cross-subject | | | | | | | Average |
|---|---|---|---|---|---|---|---|---|---|---|---|
| | SEED | SEED-IV | SHU | TUSL | HMC | TUAB | TUEV | Workload | SleepEDF | SHHS | |
| NeuroLM-B | 55.54 | 29.64 | 55.67 | 67.34 | 67.37 | 78.26 | 45.60 | 61.72 | 67.41 | 67.91 | 59.65 |
| NeuroLM-L | 60.06 | 31.98 | 56.34 | 53.14 | 66.58 | 78.76 | 41.32 | 63.11 | 67.26 | 68.42 | 58.70 |
| NeuroLM-XL | 60.34 | 32.30 | 59.36 | 68.45 | 57.61 | 79.69 | 46.79 | 63.45 | 56.40 | 59.15 | 58.35 |
| UniMind-0.5B | 66.52 | 45.54 | 64.55 | **82.56** | 74.59 | 79.87 | 60.96 | 77.33 | 71.51 | 72.08 | 69.56 |
| UniMind-1.8B | 69.05 | 44.31 | 63.79 | 80.35 | 75.08 | 80.54 | 61.64 | 77.33 | 71.79 | 73.52 | 69.74 |
| UniMind-7B | **70.55** | **45.56** | **65.77** | 78.95 | **75.27** | **81.76** | **63.19** | **78.67** | **72.98** | **74.00** | **70.67** |

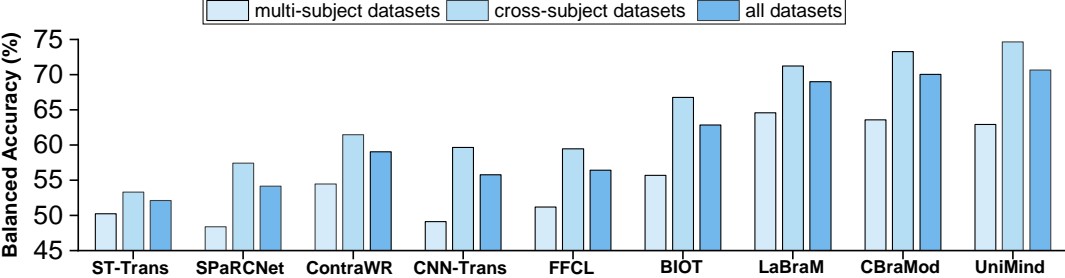

Figure 3: Performance comparison of single-task models. We report the average results of multi-subject and cross-subject settings (refer to Sec. 3.1 for datasets in each setting and **Appendices** for full results).

cross-subject and multi-subject settings. Specifically, SleepEDF, SHHS, HMC, Workload, TUAB, TUSL and TUEV adopt cross-subject setting where human subjects are non-overlappingly divided into training, validation, and test sets. The remaining datasets (SEED series and SHU) follow a multi-subject setting where training/val/test sets consists of the same cohort of subjects. We evaluate with standard metrics including Balanced Accuracy, Cohen's Kappa, and Weighted F1 (refer to **Appendices** for details).

**Implementation Details.** We utilize the EEG encoder from LaBraM (Jiang et al., 2024) and equip UniMind with three language model variants of different scales: Qwen2-0.5B (0.5B) (Yang et al., 2024), InternLM2.5-1.8B (1.8B), and InternLM2.5-7B (7B) (Cai et al., 2024). In the NLC, each branch employs an 8-head cross-attention layer, followed by a two-layer MLP. The query router $R$ is implemented as a two-layer MLP, and the query pool size is set to 16. Static queries use 1 token for both spatial and temporal dimensions, while task-adaptive queries use 2 spatial tokens and 1 temporal token. We discuss the choice of pool size and the number of queries in detail in 3.3. All experiments are conducted on a machine with eight NVIDIA A800 (80GB). Additional training details and computational resource analysis are provided in the **Appendices**.

**Compared Methods.** In the multi-task brain decoding setting, we compare *UniMind* against multi-task model baselines including NeuroLM (Jiang et al., 2025) (B, L, XL); we report the best-performing model variant for each dataset. To ensure a comprehensive evaluation, we also compare with single-task models including SPaRCNet (Jing et al., 2023), ContraWR (Yang et al., 2023b), CNN-Transformer (CNN-Trans) (Peh et al., 2022), FFCL (Li et al., 2022), ST-Transformer (ST-Trans) (Song et al., 2021), BIOT (Yang et al., 2023a), LaBraM (Jiang et al., 2024), and CBraMod (Wang et al., 2025). Notably, direct comparisons between UniMind and the baseline single-task methods are not entirely fair because those baselines are fine-tuned separately for each dataset, while UniMind uses a unified model.

## 3.2 COMPARISON WITH SOTA METHODS

We compare UniMind with both multi-task and single-task methods on ten brain decoding datasets. UniMind achieves the highest average performance, establishing itself as the leading approach for multi-task brain

Table 2: Ablation Studies (Balanced Accuracy). We report results on partial datasets due to the space limitation (refer to **Appendix** for complete results). The Average score (AVG.) is computed across all datasets.

(a) Ablation of Model Components

| Model Variant | SEED | SHHS | Workload | TUEV | AVG. |
|---|---|---|---|---|---|
| Baseline | 64.35 | 68.59 | 65.33 | 55.49 | 65.77 |
| +NLC | 69.66 | 71.18 | 70.66 | 59.79 | 68.38 |
| +NLC+TQS | **70.55** | **74.00** | **78.67** | **63.19** | **70.67** |

(b) Ablation of Using Static and Task-adaptive Queries

| Query Type | SEED | SHHS | Workload | TUEV | AVG. |
|---|---|---|---|---|---|
| Static | 69.66 | 71.18 | 70.66 | 59.79 | 68.37 |
| Adaptive | 68.51 | 73.39 | 74.00 | 62.83 | 69.34 |
| Static+Adaptive | **70.55** | **74.00** | **78.67** | **63.19** | **70.67** |

(c) The Impact of Number of Task-adaptive Queries

| $n_s$ | $n_t$ | SEED | HMC | TUAB | TUEV | AVG. |
|---|---|---|---|---|---|---|
| 1 | 2 | 66.04 | 74.71 | 79.62 | 50.15 | 66.79 |
| 1 | 2 | 68.30 | 74.00 | 80.77 | 55.41 | 68.44 |
| 2 | 1 | 68.51 | 74.62 | 80.62 | **62.83** | **69.34** |
| 2 | 2 | 68.23 | **74.67** | 80.34 | 57.89 | 68.39 |
| 2 | 4 | 67.41 | 74.06 | 80.46 | 59.27 | 67.24 |
| 4 | 2 | 67.44 | 74.11 | 80.67 | 56.70 | 67.75 |
| 4 | 4 | **68.55** | 74.10 | **81.59** | 57.38 | 68.14 |

decoding. Moreover, it demonstrates stronger cross-subject generalization than the best single-task methods. We report Balanced Accuracy here, while full metrics and error analyses are provided in the **Appendices**.

**Comparison with multi-task models.** We first compare our method with various variants of *NeuroLM* under the same multi-task setting, as shown in Table 1. *UniMind* consistently outperforms *NeuroLM* across all datasets by a large margin and demonstrates strong performance in both multi-subject and cross-subject scenarios, achieving an average improvement of **11%**. Notably, our method achieves remarkable gains on the challenging SEED-IV and TUEV datasets, substantially improving their initially low balanced accuracies (*i.e.,* ↑**13.26%** and ↑**16.4%**), thanks to the NLC, which enables the LLM to better comprehend rich temporal and spatial semantic cues in EEG signals. Besides, *UniMind* achieves a **15.22%** increase in balanced accuracy on the small-scale Workload dataset with only 1k samples. These improvements stem from TQS's task-adaptive query selection from a shared pool, which facilitates knowledge transfer and mutual enhancement among tasks. Moreover, our model exhibits stable performance across parameters. Larger models benefit challenging datasets like SEED and TUEV, whereas smaller models excel on limited datasets like TUSL.

**Comparison with single-task models.** We further compare *UniMind* with state-of-the-art single-task models. It should be noted that multi-task learning is inherently more challenging. Therefore, this comparison is not entirely fair and is intended primarily as a reference for performance evaluation. As shown in Figure 3, although *UniMind* performs slightly below the best single-task model in the multi-subject setting, it achieves the best results in the cross-subject setting. Moreover, UniMind also obtains the **best** average score across all datasets with a single model, demonstrating its superior potential for multi-task decoding, outperforming most single-task baselines without the need for additional task-specific tuning.

## 3.3 ABLATION STUDY

**Analysis of Key Model Components.** We conduct an ablation study to evaluate the contributions of key model components, as shown in Table 2a. The baseline model employs a fixed number of learnable queries to extract features from the entire EEG sequence. Introducing the NLC module results in substantial performance gains across all datasets, with improvements around 5% on more challenging tasks such as SEED and TUEV. This is due to the NLC's ability to capture temporal and spatial patterns in EEG embeddings and convert them into LLM's semantic representations. The addition of the TQS further enhances performance, yielding gains of 3.40% on TUEV, 2.82% on SHHS, and 8.01% on Workload. These improvements stem from the TQS that enables adaptive query learning across tasks, promoting knowledge sharing among related tasks while reducing inter-task interference in cross-modality bridging.

Table 3: Negative transfer analysis. Naive joint training leads to performance drops without TQS.

| Training Strategy | TUEV | TUAB |
|---|---|---|
| Single-Task Baseline | **64.09** | **81.40** |
| Naive Joint Training | 37.91 | 74.92 |

Table 4: Generative vs. Contrastive. UniMind consistently outperforms both baselines.

| Dataset | LaBraM | Contrastive | UniMind |
|---|---|---|---|
| HMC | 72.86 | 72.16 | **75.27** |
| TUAB | 81.40 | 80.24 | **81.76** |

**Comparison of Different Types of Queries.** We compare different query types used in the NLC module to evaluate the impact of static and task-adaptive query selection, as shown in Table 9. Compared to using only static queries, employing task-adaptive queries selected by the TQS module yields notable improvements on most datasets, with gains of 3.04% on TUEV and 3.34% on Workload. This improvement is due to the ability of each task to adaptively select queries that better capture task-relevant temporal and spatial features. Furthermore, combining static and task-adaptive queries yields the best performance across all datasets.

**Robustness to LLM Selection.** We investigate the impact of different language backbones on model performance. Specifically, the 0.5B variant employs Qwen2-0.5B (as the InternLM2.5 series begins at 1.8B), while the 1.8B and 7B variants are built upon InternLM2.5. As shown in Table 1, the consistent scaling performance observed across different LLM architectures suggests that UniMind is robust to the specific choice of LLM architecture. Additionally, replacing the GPT-2 backbone in NeuroLM with InternLM2.5-1.8B on TUAB yielded a balanced accuracy of 78.49%, comparable to the original 79.69%. This confirms that merely upgrading the backbone has a minimal impact on the performance of existing methods, further highlighting the effectiveness of our architecture regardless of the specific LLM.

**Analysis of Hyperparameters.** We investigate several key hyperparameters. First, the number of task-adaptive queries ($n_s$ for $q_a^s$ and $n_t$ for $q_a^t$) influences performance: too few queries limit the model's capacity to capture diverse EEG patterns, while too many may introduce noise. As shown in Table 2c, $n_s = 2$ and $n_t = 1$ yield the best results. Besides, the query pool size ($\boldsymbol{P}_t$ and $\boldsymbol{P}_s$, denoted $n_q$) affects task-specific selection and inter-task interactions. Small pools lead to overlapping selections, whereas overly large pools weaken mutual enhancement; Figure 4 shows that a pool size of 16 balances diversity and shared representation.

**Analysis of Task Synergy and Data Scaling.** To disentangle the contributions of architectural design from simple data scaling, we investigated whether the multi-task performance gains could be achieved by merely increasing data volume without effective architectural design. We designed a controlled experiment where the LaBraM baseline was fine-tuned jointly on the TUAB and TUEV datasets. However, as shown in Table 3, this naive mixing strategy led to severe negative transfer, with balanced accuracy dropping to 37.91% on TUEV (from 64.09%) and to 74.92% on TUAB (from 81.40%). This outcome decisively confirms that simply aggregating datasets is insufficient for multi-task learning in EEG; instead, a task-aware mechanism like our TQS module is essential to mitigate interference and achieve positive synergy.

## 3.4 COMPARISON WITH CONTRASTIVE ALIGNMENT PARADIGM.

Contrastive learning methods typically align EEG and text modalities by optimizing their similarity in a shared latent space. To validate the superiority of our generative architecture over contrastive learning methods, we implemented a contrastive baseline inspired by recent works like EEG-CLIP Ndir et al. (2025). Specifically, the LaBraM encoder is fully fine-tuned end-to-end to align EEG representations with pre-extracted text embeddings, optimized via a SoftCLIP loss Gao et al. (2023).

The quantitative results are presented in Table 4, showing that jointly contrastively tuning LaBraM on both HMC and TUAB leads to a performance drop compared to training on each dataset individually, indicating

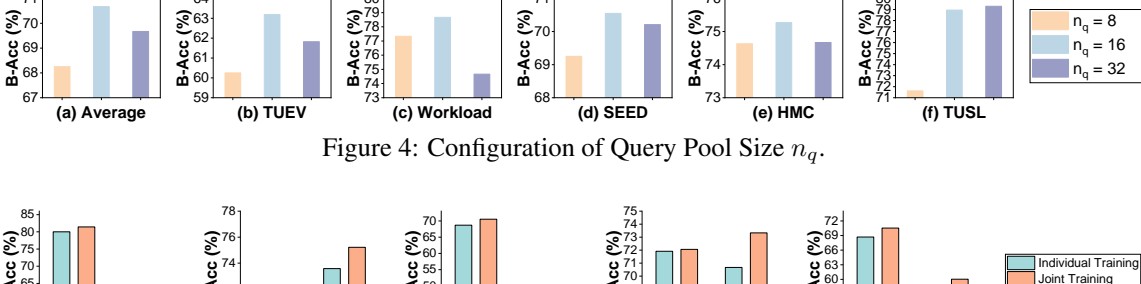

Figure 4: Configuration of Query Pool Size $n_q$.

Figure 5: Comparison of individual and joint training balanced accuracy across multiple tasks.

that the multi-task setting is upper-bounded by the single-task performance. We attribute this to the absence of a task-aware reasoning ability, which limits the model's ability to handle task heterogeneity. Forcing a shared encoder to accommodate different tasks can dilute the quality of the learned EEG representations. In contrast, UniMind learns task-adaptive query tokens that inject explicit task awareness into the rich spatiotemporal semantic cues distilled by NLC, unleashing LLM's complex reasoning ability in decoding multi-task EEG signals. Furthermore, unlike contrastive methods limited to retrieval, our generative framework inherently supports open-ended question answering, enabling more flexible and comprehensive brain signal decoding.

### 3.5 TASK-LEVEL QUERY ANALYSIS

In this section, we explore task-level query selection patterns and inter-task correlations, and further investigate the mutual enhancement effect, examining whether functionally similar tasks benefit from joint training.

**Visualization of Task-adaptive Query Selection.** To investigate the working mechanism of the TQS module, we visualize the spatial and channel routing scores $S$ from query routers $R$ across EEG datasets using t-SNE. As shown in Figure 6 (a), the neural routing distribution presents a clear **task-aware pattern** for both temporal and spatial routers, with samples from different datasets forming clearly separated clusters. It suggests that TQS is capable of dynamically selecting queries across heterogeneous tasks.

Figure 6 (b) shows the selection frequencies of task-adaptive spatial queries across datasets, indicating clear task-dependent trends. For example, emotion recognition and clinical event detection tasks tend to select queries $P_4$ and $P_{10}$, while sleep stage classification tasks consistently favor $P_5$. In addition, Figure 7 (b) illustrates the channels attended by different queries in the spatial query pool, revealing noticeable biases. Figure 7 (c) further aggregates the preferred queries for each task and their channel-wise attention, providing a topographical visualization of task-specific channel focus.

**Neural Correlations across Tasks.** To capture underlying neural correlations across tasks, we measure similarities in task-adaptive query distributions among EEG tasks to reflect neural patterns. As illustrated in Figure 7 (a), we highlight two key observations. (1) Datasets/tasks belonging to the same task domain tend to exhibit much higher similarity scores. For example, TUAB and TUEV (clinical) show remarkably consistent selection patterns. These findings indicate that TQS can help capture consistent neural activation patterns from similar tasks. (2) Certain tasks from different domains exhibit strong similarities, suggesting shared neural mechanisms across cognitive functions. For example, clinical anomaly detection tasks (TUAB, TUEV) and emotion recognition tasks (SEED, SEED-IV) show similar spatial patterns, due to overlapping activity in the bilateral temporal regions (Zheng & Lu, 2015; Rahman et al., 2021), particularly around electrodes T7 and T8. These findings reveal underlying neural correlations between tasks, which is consistent with existing neuroscientific research, implying the potential of shared representations for multi-task learning.

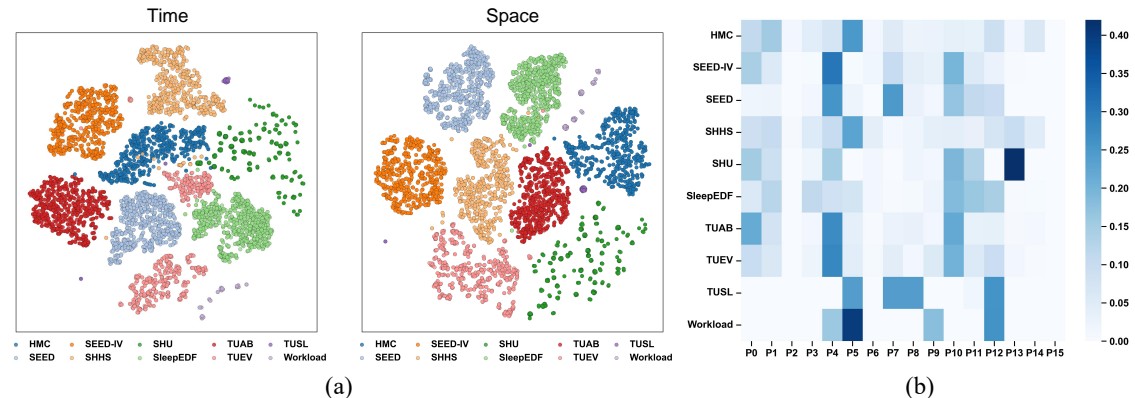

(a)  (b)

Figure 6: (a): t-SNE-based visualization of neural routing distributions across datasets; (b): task-adaptive spatial query distributions across datasets.

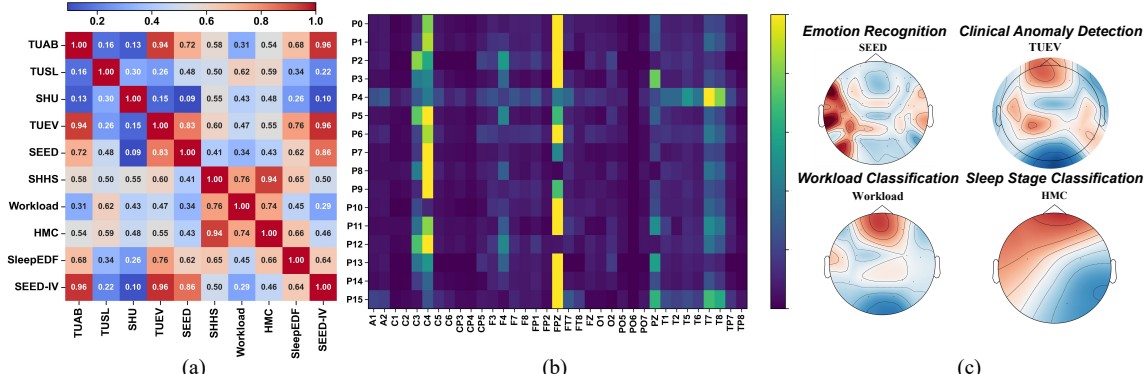

(a)  (b)  (c)

Figure 7: (a): Similarity of task-adaptive query distributions across tasks; (b): The attention values of task-adaptive queries across different channels; (c): Topography visualization on different tasks.

**Mutual Enhancement in Multi-task Training.** We further validate the mutual enhancement effect in multi-task training by selecting six pairs of datasets with high similarity scores. As shown in Figure 5, joint training consistently leads to better performance than training each task separately. The results demonstrate that UniMind facilitates effective multi-task training, enhancing the performance across diverse decoding tasks by sharing neural representations.

## 4 CONCLUSION

In this paper, we present UniMind, a general-purpose EEG foundation model for multi-task brain decoding. It leverages a Neuro-Language Connector to align spatiotemporal EEG features with LLMs and a Task-aware Query Selection Module to adapt to diverse tasks. Experiments on ten datasets show UniMind outperforms prior models. We believe this work lays a solid foundation for future research on LLM–brain interaction and multi-task learning in the EEG domain.

## REPRODUCIBILITY STATEMENT

We provide detailed descriptions of our model training settings and dataset construction in the appendix, covering batch size, learning rate schedules, and optimizer configurations. The backbone model Cai et al. (2024) is publicly available, and all external datasets used in our work are either publicly released or cited with appropriate references.

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

## A APPENDIX

### A.1 LLM USAGE STATEMENT

In this study and paper drafting, large language models (LLMs) are only used for statement polishing. Specifically, their application focuses on optimizing text expression, such as refining the logical coherence of descriptions, adjusting the accuracy of academic terms, and enhancing the readability of experimental results. LLMs are not involved in core research links, including data collection, experimental design, model construction, result analysis, and conclusion derivation. All content polished by LLMs has been strictly reviewed by the research team to ensure consistency with original research intentions and academic authenticity, without introducing false or misleading information.

### A.2 RELATED WORKS

**Task-Specific EEG Decoding Models.** Due to the variations in EEG signal formats across different datasets, numerous deep learning models have been proposed to tackle the aforementioned tasks within their respective domains. These models primarily focus on designing tailored feature extraction networks to accommodate EEG samples for specific tasks. Some works Jing et al. (2023); Nagabushanam et al. (2020); Dar et al. (2020) apply 1D convolutional neural networks (CNNs) directly to raw signals, while others Yang et al. (2022); Kim et al. (2020) first preprocess the data using short-time Fourier transform (STFT) and then adopt 2D CNNs on the resulting spectrograms. However, these methods offer a relatively narrow scope of perception, focusing mainly on local information. To address this, some methods Jing et al. (2020); Almutairi et al. (2021); Yang et al. (2023b); Li et al. (2022) build on CNN-based encoders by incorporating Transformers, LSTMs, or recurrent neural networks (RNNs) to model temporal dynamics and capture global dependencies. To further enhance EEG feature representation, some approaches employ multiple encoders for ensemble learning Li et al. (2022), or leverage multi-level Transformers to encode spatial and temporal features both across and within channelsSong et al. (2021); Liu et al. (2022). However, these task-specific models often suffer from limited generalization and poor transferability, restricting their applicability to broader EEG analysis tasks.

**EEG Foundation Model.** Because task-specific models can only handle a single task and lack cross-task learning capabilities, their broader applicability is limited. As a result, EEG foundation models have attracted increasing attention. These models aim to learn universal EEG representations by training on diverse datasets, enabling generalization across various downstream tasks. To address the challenges posed by heterogeneous EEG formats, Yang et al. Yang et al. (2023a) proposed BIOT, which tokenizes EEG channels into fixed-length segments and incorporates channel and relative position embeddings to preserve spatio-temporal features. Inspired by masked language modeling in LLMs, Jiang et al. Jiang et al. (2024) extended this idea with LaBraM, introducing a neural tokenizer and pre-training it via masked neural code prediction on large-scale EEG data, achieving SOTA results across multiple tasks. While these models effectively address many challenges, they still require separate fine-tuning for each downstream task.

Recently, contrastive learning paradigms like CLIP Ndir et al. (2025) and ELM-MIL Gijsen & Ritter (2025) have been explored to align EEG signals with natural language by optimizing a shared embedding space. However, such discriminative approaches are strictly upper-bounded by the representational capacity of the EEG encoder; since the encoder alone must encapsulate all semantic information to match the text embedding, any information loss in the encoder leads to an irreversible performance bottleneck. Furthermore, these methods are primarily designed for retrieval tasks, lacking the inherent capability to support open-ended question answering and complex reasoning. To harness the cross-modal understanding potential power of generative models, NeuroLM Jiang et al. (2025) was introduced as the first multi-task foundation model for EEG, leveraging the capabilities of LLMs to support multi-task learning and inference. However, its performance is limited by the modality gap between EEG and language, as well as interference among tasks. In contrast, UniMind bridges this gap through a spatio-temporal cross-modality alignment strategy and

reduces task interference via task-aware query selection. This design allows UniMind to achieve stronger performance across diverse tasks while supporting unified multi-task decoding within a single model.

**Multimodal Large Language Models.** Building on the success of Large Language Models (LLMs), the Multimodal Large Language Models (MLLMs) are developed to enhance cross-modal understanding by combining visual, auditory, and textual data. Some apporaches Alayrac et al. (2022); Awadalla et al. (2023); Li et al. (2023); Zhang et al. (2023b) enhance LLMs by incorporating components like gated cross-attention layers or adapter layers to handle multimodal inputs. Others Dai et al. (2023); Liu et al. (2023); Zhu et al. (2023) use projection layers or Q-Formers to map visual encoder outputs into LLM input space. Video LLMs Li et al. (2024); Zhang et al. (2023a); Maaz et al. (2024); Luo et al. (2023); Ji et al. (2024) extend MLLMs for video tasks, primarily using projection layers or Q-Formers to process visual tokens. Besides, recent MLLMs have successfully integrated modalities such as audio Cheng et al. (2024); Su et al. (2023); Wu et al. (2024), applying them to EEG signals remains particularly challenging. While both audio and EEG exhibit continuous temporal patterns, EEG is inherently more complex due to its non-stationary characteristics across both temporal and spatial dimensions. It captures fine-grained neural activity that varies over time and across different brain regions, introducing significant variability and noise. Unlike text, which is composed of discrete and structured tokens, EEG signals are fluid, high-dimensional, and constantly changing. This makes it difficult for LLMs to directly process and align with EEG representations, posing a major challenge for effective cross-modal integration. Building on these insights, UniMind proposes the Neuro-Language Connector, a crucial module that bridges the substantial modality gap between continuous, complex EEG signals and the discrete token representations of LLMs. By encoding spatiotemporal neural patterns into interpretable features, it enables more effective cross-modal alignment, paving the way for improved integration of EEG within multimodal language models.

### A.3 DATASET DETAILS

**Dataset information.** To comprehensively evaluate *UniMind*, we utilize ten publicly available EEG datasets covering five representative task domains. (1) For **sleep stage classification**, we employ HMC Alvarez-Estevez & Rijsman (2021), SleepEDF Kemp et al. (2000), and SHHS Quan et al. (1997). These datasets provide long-term polysomnographic EEG recordings annotated by experts, with each EEG segment labeled into one of five standard sleep stages: Wake, NREM-1, NREM-2, NREM-3, and REM. (2) For **emotion recognition**, we adopt SEED Zheng & Lu (2015) and SEED-IV Zheng et al. (2019), both containing multi-session EEG data collected while subjects viewed emotional video stimuli. SEED includes three emotion categories (positive, negative, neutral), whereas SEED-IV expands this to four categories: neutral, sad, fear, and happy. (3) For **Clinical Abnormaly Detection**, we use TUAB Harati et al. (2015), TUEV Harati et al. (2015), and TUSL von Weltin et al. (2017). TUAB focuses on detecting abnormal EEG activity, TUEV classifies six types of clinical events such as epileptiform discharges and eye movements, and TUSL categorizes seizure, slowing, and complex background activity. (4) For **cognitive workload classification**, we utilize the Workload dataset Zyma et al. (2019), which contains EEG recordings collected during mental arithmetic tasks designed to distinguish between high and low cognitive load. (5) Finally, for **motor imagery classification**, we incorporate SHU Ma et al. (2022), a large-scale dataset featuring EEG signals corresponding to imagined left- and right-hand movements, enabling binary decoding of motor intentions in brain–computer interface applications.

**Data Splits.** Each dataset is divided into training, validation, and test sets according to task-specific strategies. (1) **TUAB and TUEV**: we follow the official training and test split, and further divide the training subjects into 80% for training and 20% for validation. (2) **SEED and SEED-IV**: trials are split chronologically. SEED contains 15 trials, divided into 9 for training, 3 for validation, and 3 for testing; SEED-IV contains 24 trials, divided into 16, 4, and 4 respectively. All sessions within each split are merged. (3) **HMC**: subjects 1 to 100 are assigned to the training set, 101 to 126 to the validation set, and 127 to 154 to the test set. (4) **Workload**: subjects 0 to 25 are used for training, 26 to 30 for validation, and 31 to 35 for testing. (5) **TUSL**: the dataset is

Table 5: Overview of Downstream EEG datasets and tasks used for evaluation.

| Task | Dataset | Rate (Hz) | #Channels | Duration | #Samples | #Subjects | Label |
|---|---|---|---|---|---|---|---|
| Motor Imagery | SHU Ma et al. (2022) | 250 | 32 | 4s | 11,988 | 25 | Binary-class |
| Emotion Recognition | SEED Zheng & Lu (2015) | 1000 | 62 | 1s | 144,851 | 15 | 3-class |
| | SEED-IV Zheng et al. (2019) | 1000 | 62 | 1s | 143,610 | 16 | 5-class |
| Clinical Abnormaly Detection | TUAB Harati et al. (2015) | 250/256/512 | 23 | 10s | 409,083 | 2,383 | Binary-class |
| | TUEV Harati et al. (2015) | 250 | 23 | 5s | 112,237 | 370 | 6-class |
| | TUSL von Weltin et al. (2017) | 250 | 23 | 10s | 245 | 28 | 3-class |
| Sleep Classification | SHHS Quan et al. (1997) | 125 | 1 | 30s | 324,854 | 329 | 5-class |
| | SleepEDF Kemp et al. (2000) | 100 | 1 | 30s | 414,961 | 78 | 5-class |
| | HMC Alvarez-Estevez & Rijsman (2021) | 256 | 4 | 30s | 137,243 | 151 | 4-class |
| Workload Classification | Workload Zyma et al. (2019) | 500 | 19 | 4s | 1080 | 36 | Binary-class |

randomly divided into 60% for training, 20% for validation, and 20% for testing. (6) **SleepEDF and SHHS**: subjects are randomly assigned to training, validation, and test sets in an 8:1:1 ratio. (7) **SHU**: sessions are split into training, validation, and test sets in a 3:1:1 ratio.

**Data Preprocessing.** We develop a standardized data pre-processing pipeline comprising band-pass filtering, notch filtering, resampling, and normalization, designed to be universally applicable across various BCI tasks and decoding models. Following established practice, we apply a 0.1–75 Hz band-pass filter to retain task-relevant frequency components while suppressing low-frequency drifts and high-frequency noise. To remove power-line interference, we first perform a Fast Fourier Transform (FFT) on the raw EEG data to identify the specific interference frequency, and then apply a notch filter at 50 Hz or 60 Hz based on the local power-line frequency. Finally, all EEG signals are resampled to 200 Hz. To mitigate the effects of low-amplitude signals and enhance numerical stability during optimization, we apply z-score normalization to most datasets. Specifically, for the SHU dataset, we instead use 95% normalization due to its signal distribution characteristics.

**Metrics.** Due to the inherent class imbalance in EEG datasets, we evaluate model performance using three primary metrics. **(1) Balanced Accuracy** is defined as the average recall obtained on each class. By equally weighting the recall of all classes, it effectively mitigates the bias caused by imbalanced class distributions, ensuring that minority classes are not overlooked during evaluation. **(2) Cohen's Kappa** measures the level of agreement between the predicted labels and the true labels, while correcting for the agreement that could occur by random chance. This statistic provides a more robust assessment of classification performance than simple accuracy, especially in datasets where class distribution is uneven. **(3) Weighted F1 Score** is the harmonic mean of precision and recall calculated for each class individually and then averaged by weighting each class's score according to its support (i.e., the number of true instances). This metric balances false positives and false negatives while accounting for class imbalance, providing a comprehensive measure of overall model effectiveness.

Since our model produces discrete class labels directly from LLM outputs without associated probability scores, metrics that require confidence values such as **AUROC** (Area Under the Receiver Operating Characteristic Curve) and **AUC-PR** (Area Under the Precision-Recall Curve) are not applicable. Therefore, **Weighted F1** and **Cohen's Kappa** are prioritized for both binary and multi-class classification tasks to ensure reliable evaluation under these constraints.

## A.4 TRAINING DETAILS

**Experimental Setup.** All experiments are conducted on a single machine equipped with eight NVIDIA A800 GPUs, each with 80GB of memory. We adopt a batch-wise alternating training strategy, where each mini-batch is drawn from a single dataset. The software environment includes Python 3.9 and PyTorch 2.0.1 with CUDA 11.8 support. We train our model using 8-GPU data parallelism with a total effective batch size

of 128. Training is performed for 10 epochs using the AdamW optimizer with a cosine learning rate schedule, a base learning rate of 4e-5, and a warmup ratio of 3%. Please refer to Table 6 for additional training details.

Table 6: Key training and model configuration parameters

| Parameter | Value |
|---|---|
| Training Epochs | 10 |
| Batch Size | 128 |
| Learning Rate | $4 \times 10^{-5}$ |
| Weight Decay | 0.01 |
| Warmup Ratio | 0.03 |
| LR Scheduler | Cosine decay |
| Max Sequence Length | 4096 |
| Dataloader Workers | 4 |
| BFloat16 Precision | True |
| Use Thumbnail | True |
| EEG Hidden Size | 1152 |
| Number of EEG Encoder Layers | 12 |
| Number of EEG Attention Heads | 10 |
| LLM Hidden Size | 4096 |
| Number of LLM Layers | 32 |
| Number of LLM Attention Heads | 32 |
| RoPE Scaling | Dynamic, factor = 2.0 |

**Computational Requirements.** Our language models adopt three different sizes, corresponding to Qwen2-0.5B, InternLM-1.8B, and InternLM-7B as shown in Table 7. The table reports the training time and maximum GPU memory consumption on 8 NVIDIA A800 GPUs.

Table 7: Training resource comparison of UniMind

| Language Model Size | Time (8-GPU, hours) | Max GPU Memory (GB) |
|---|---|---|
| Qwen2-0.5B | 7.34 | 31.2 |
| InternLM-1.8B | 8.37 | 39.4 |
| InternLM-7B | 21.78 | 61.7 |

## A.5 MORE ABLATION STUDIES

In this section, we provide detailed ablation results that were condensed in the main text due to space limitations, offering a more comprehensive analysis of key hyperparameters and module components.

**Analysis of Key Model Components.** Table 8 presents the full breakdown of performance gains from introducing the NLC and TQS modules. The results confirm that the combination of NLC and TQS consistently yields the best performance across all datasets.

**Comparison of Different Types of Queries.** Table 9 compares different query types used in the NLC module to evaluate the impact of static and task-adaptive query selection. The comprehensive results validate that combining static and task-adaptive queries yields the best performance across all datasets.

Table 8: Ablation of Model Components (Balanced Accuracy)

| Model Variant | AVG | SEED | HMC | TUAB | TUEV | Workload | TUSL | SEED-IV | SleepEDF | SHHS | SHU |
|---|---|---|---|---|---|---|---|---|---|---|---|
| Baseline | 65.77 | 64.35 | 73.87 | 79.63 | 55.49 | 65.33 | 72.65 | 42.74 | 71.07 | 68.59 | 63.96 |
| +NLC | 68.38 | 69.66 | 74.43 | 80.97 | 59.79 | 70.66 | 76.69 | 43.40 | 72.01 | 71.18 | 64.90 |
| +NLC+TQS | **70.67** | **70.55** | **75.27** | **81.76** | **63.19** | **78.67** | **78.95** | **45.56** | **72.98** | **74.00** | **65.77** |

Table 9: Ablation of Using Static and Task-adaptive Queries (Balanced Accuracy)

| Query Type | AVG | SEED | HMC | TUAB | TUEV | Workload | TUSL | SEED-IV | SleepEDF | SHHS | SHU |
|---|---|---|---|---|---|---|---|---|---|---|---|
| Static | 68.37 | 69.66 | 74.43 | 80.97 | 59.79 | 70.66 | 76.69 | 43.40 | 72.01 | 71.18 | 64.90 |
| Adaptive | 69.34 | 68.51 | 74.62 | 80.62 | 62.83 | 74.00 | 77.07 | 44.77 | 72.26 | 73.39 | 65.30 |
| Static+Adaptive | **70.67** | **70.55** | **75.27** | **81.76** | **63.19** | **78.67** | **78.95** | **45.56** | **72.98** | **74.00** | **65.77** |

Table 10: The Impact of Number of Task-adaptive Queries (Balanced Accuracy)

| $n_s$ | $n_t$ | AVG | SEED | HMC | TUAB | TUEV | Workload | TUSL | SEED-IV | SleepEDF | SHHS | SHU |
|---|---|---|---|---|---|---|---|---|---|---|---|---|
| 1 | 2 | 66.79 | 66.04 | 74.71 | 79.62 | 50.15 | 76.00 | 74.77 | 43.64 | 69.87 | 69.37 | 63.76 |
| 1 | 2 | 68.44 | 68.30 | 74.00 | 80.77 | 55.41 | **78.67** | **77.95** | 44.05 | 70.51 | 70.64 | 64.14 |
| 2 | 1 | **69.34** | 68.51 | 74.62 | 80.62 | **62.83** | 74.00 | 77.07 | **44.77** | 72.26 | **73.39** | **65.30** |
| 2 | 2 | 68.39 | 68.23 | **74.67** | 80.34 | 57.89 | 73.18 | 77.42 | 44.36 | 72.07 | 72.50 | 63.27 |
| 2 | 4 | 67.24 | 67.41 | 74.06 | 80.46 | 59.27 | 74.66 | 65.86 | 43.50 | **72.98** | 71.69 | 62.47 |
| 4 | 2 | 67.75 | 67.44 | 74.11 | 80.67 | 56.70 | 71.33 | 77.10 | 44.51 | 71.39 | 71.52 | 62.76 |
| 4 | 4 | 68.14 | **68.55** | 74.10 | **81.59** | 57.38 | 74.67 | 75.02 | 44.10 | 71.62 | 71.58 | 62.80 |

**Analysis of the Number of Task-adaptive Queries.** Table 10 details the impact of the number of task-adaptive queries in Table 10, where $n_s$ and $n_t$ denote the numbers of task-adaptive queries $q_a^s$ and $q_a^t$, respectively. The configuration $n_s = 2, n_t = 1$ achieves the highest average balanced accuracy.

**Impact of Query Pool Size** ($n_q$). Table 11 shows the performance across different query pool sizes. The full dataset analysis confirms that $n_q = 16$ provides the optimal balance between diversity and shared representation for the majority of tasks.

**Efficacy in Alleviating Data Scarcity.** To investigate whether UniMind can effectively alleviate the data scarcity problem, we analyzed the performance gains obtained by training on massive multi-task data compared to single-task baselines. As shown in Table 12, on the TUSL dataset (containing only 245 samples), joint training boosts balanced accuracy by a remarkable **+28.81%** (rising from 50.14% to 78.95%). Similarly, on the Workload dataset (1,080 samples), performance improves by **+8.34%** (from 70.33% to 78.67%). These substantial improvements confirm that UniMind effectively mitigates data scarcity by leveraging shared representations learned from diverse tasks.

Table 11: **Impact of Query Pool Size** ($n_q$) **on All 10 Datasets.** The best results are highlighted in bold. (Metric: Balanced Accuracy %)

| $n_q$ | AVG | SEED | HMC | TUAB | TUEV | Workload | TUSL | SEED-IV | SleepEDF | SHHS | SHU |
|---|---|---|---|---|---|---|---|---|---|---|---|
| 8 | 68.25 | 69.25 | 74.63 | 79.87 | 60.25 | 77.33 | 71.62 | 43.89 | 71.51 | 72.53 | 61.59 |
| **16** | **70.67** | **70.55** | **75.27** | **81.76** | **63.19** | **78.67** | 78.95 | **45.56** | **72.98** | **74.00** | **65.77** |
| 32 | 69.67 | 70.21 | 74.67 | 80.81 | 61.82 | 74.66 | **79.29** | 45.17 | 72.10 | 72.95 | 65.04 |

Table 12: Performance Comparison between Single-Task and Joint Training on UniMind of Balanced Accuracy.

| Strategy | SEED | HMC | TUAB | TUEV | TUSL | Workload | SEED-IV | SleepEDF | SHHS | SHU |
|---|---|---|---|---|---|---|---|---|---|---|
| Single-Task | 69.79 | 74.35 | 80.22 | 52.81 | 50.14 | 70.33 | 39.06 | 71.09 | 72.91 | 65.26 |
| Joint Training | 70.55 | 75.27 | 81.76 | 63.19 | 78.95 | 78.67 | 45.56 | 72.98 | 74.00 | 65.77 |

## A.6 COMPLETE PERFORMANCE COMPARISON ON ALL METRICS

We present the complete results for Balanced Accuracy, Cohen's Kappa, and Weighted F1 across all ten datasets in Tables 13–17. By conducting multiple experiments with different random seeds, we calculate the mean values along with their standard deviations. The outcomes on these three metrics are generally consistent with the analysis provided in the main text. Additionally, UniMind demonstrates relatively smaller standard deviations compared to other models, especially on the smaller datasets such as TUSL and Workload, which indicates more stable and reliable performance.

Table 13: Performance comparison on SEED Zheng & Lu (2015) and HMC Alvarez-Estevez & Rijsman (2021) datasets. "MT" denotes multi-task learning usage. Underlining shows best multi-task results; **bold** shows best overall.

| Model | MT | SEED | | | HMC | | |
|---|---|---|---|---|---|---|---|
| | | B-Acc | Kappa | F1-W | B-Acc | Kappa | F1-W |
| SPaRCNet | ✗ | 55.96 ($\pm$2.44) | 34.64 ($\pm$3.72) | 55.85 ($\pm$2.97) | 47.56 ($\pm$11.09) | 31.47 ($\pm$13.15) | 41.08 ($\pm$13.10) |
| ContraWR | ✗ | 61.06 ($\pm$0.78) | 42.20 ($\pm$1.29) | 61.37 ($\pm$0.85) | 42.42 ($\pm$5.54) | 23.40 ($\pm$5.54) | 29.87 ($\pm$2.88) |
| CNN-Trans. | ✗ | 61.61 ($\pm$3.84) | 42.62 ($\pm$6.01) | 61.50 ($\pm$4.63) | 65.73 ($\pm$1.41) | 59.61 ($\pm$1.05) | 68.96 ($\pm$0.65) |
| FFCL | ✗ | 58.08 ($\pm$3.22) | 37.32 ($\pm$4.62) | 57.43 ($\pm$4.02) | 44.27 ($\pm$7.02) | 25.42 ($\pm$6.54) | 29.02 ($\pm$4.85) |
| ST-Trans. | ✗ | 54.79 ($\pm$0.91) | 32.61 ($\pm$1.69) | 55.05 ($\pm$0.91) | 25.59 ($\pm$1.41) | 5.03 ($\pm$1.83) | 14.28 ($\pm$1.22) |
| BIOT | ✗ | 70.97 ($\pm$0.24) | 56.82 ($\pm$0.51) | 71.34 ($\pm$0.27) | 68.62 ($\pm$0.41) | 62.95 ($\pm$1.13) | 70.91 ($\pm$1.47) |
| LaBraM | ✗ | **73.18** ($\pm$0.19) | **59.94** ($\pm$0.31) | **73.54** ($\pm$0.21) | 72.86 ($\pm$1.01) | 68.12 ($\pm$0.73) | 75.54 ($\pm$0.24) |
| CBraMod | ✗ | 65.94 ($\pm$0.19) | 49.24 ($\pm$0.17) | 66.41 ($\pm$0.19) | 69.46 ($\pm$0.16) | 63.43 ($\pm$0.15) | 71.43 ($\pm$0.17) |
| NeuroLM | ✓ | 60.34 ($\pm$0.10) | 40.82 ($\pm$0.36) | 60.63 ($\pm$0.30) | 67.37 ($\pm$0.50) | 61.88 ($\pm$0.57) | 71.26 ($\pm$0.34) |
| UniMind | ✓ | 70.55 ($\pm$0.61) | 56.28 ($\pm$0.73) | 70.98 ($\pm$0.96) | **75.27** ($\pm$0.56) | **70.58** ($\pm$0.48) | **77.40** ($\pm$0.73) |

Table 14: Performance comparison on TUAB Harati et al. (2015) and TUEV Harati et al. (2015) datasets. "MT" denotes multi-task learning usage. Underlining shows best multi-task results; **bold** shows best overall.

| Model | MT | TUAB | | | TUEV | | |
|---|---|---|---|---|---|---|---|
| | | B-Acc | Kappa | F1-W | B-Acc | Kappa | F1-W |
| SPaRCNet | ✗ | 78.69 ($\pm$0.47) | 50.67 ($\pm$0.29) | 75.13 ($\pm$0.80) | 41.61 ($\pm$2.62) | 42.33 ($\pm$1.81) | 70.24 ($\pm$1.04) |
| ContraWR | ✗ | 80.17 ($\pm$0.58) | 61.02 ($\pm$0.38) | 80.65 ($\pm$0.42) | 43.84 ($\pm$3.49) | 39.12 ($\pm$2.37) | 68.93 ($\pm$1.36) |
| CNN-Trans. | ✗ | 79.53 ($\pm$0.94) | 57.33 ($\pm$0.63) | 78.76 ($\pm$0.31) | 40.87 ($\pm$1.61) | 38.15 ($\pm$1.34) | 68.54 ($\pm$2.93) |
| FFCL | ✗ | 78.19 ($\pm$0.22) | 55.81 ($\pm$0.84) | 77.83 ($\pm$0.75) | 39.79 ($\pm$1.04) | 37.32 ($\pm$1.88) | 67.83 ($\pm$1.20) |
| ST-Trans. | ✗ | 79.66 ($\pm$0.31) | 61.69 ($\pm$0.45) | 80.90 ($\pm$0.93) | 39.84 ($\pm$2.28) | 37.65 ($\pm$3.06) | 68.23 ($\pm$1.90) |
| BIOT | ✗ | 79.59 ($\pm$0.89) | 59.42 ($\pm$0.13) | 78.82 ($\pm$0.90) | 52.81 ($\pm$2.25) | 52.73 ($\pm$2.49) | 74.92 ($\pm$0.82) |
| LaBraM | ✗ | 81.40 ($\pm$0.20) | 62.52 ($\pm$0.12) | 81.47 ($\pm$0.57) | **64.09** ($\pm$0.65) | **66.37** ($\pm$0.93) | **83.12** ($\pm$0.52) |
| CBraMod | ✗ | 79.79 ($\pm$0.22) | 63.02 ($\pm$0.28) | **85.29** ($\pm$0.26) | 40.23 ($\pm$0.22) | 39.15 ($\pm$0.27) | 69.73 ($\pm$0.25) |
| NeuroLM | ✓ | 79.69 ($\pm$0.53) | 54.61 ($\pm$0.98) | 78.93 ($\pm$0.91) | 46.79 ($\pm$3.56) | 45.70 ($\pm$4.98) | 73.59 ($\pm$2.19) |
| UniMind | ✓ | **81.76** ($\pm$0.33) | **63.80** ($\pm$0.77) | 82.03 ($\pm$0.60) | 63.19 ($\pm$1.93) | 56.29 ($\pm$1.11) | 78.04 ($\pm$0.84) |

## A.7 INSTRUCTION CONSTRUCTION

We adopt instruction tuning to train UniMind. To enhance instruction diversity, we design ten distinct prompts for each dataset, with each sample paired with one randomly selected task-specific instruction. Notably, UniMind is robust to prompt variations: training with a single fixed prompt yields results statistically identical

Table 15: Performance comparison on TUSL Zheng et al. (2019) and Workload Kemp et al. (2000) datasets. "MT" denotes multi-task learning usage. Underlining shows best multi-task results; **bold** shows best overall.

| Model | MT | TUSL | | | Workload | | |
|---|---|---|---|---|---|---|---|
| | | B-Acc | Kappa | F1-W | B-Acc | Kappa | F1-W |
| SPaRCNet | ✗ | 41.85 (±4.52) | 13.99 (±7.99) | 35.00 (±9.68) | 59.77 (±3.67) | 10.67 (±2.42) | 54.60 (±4.11) |
| ContraWR | ✗ | 58.57 (±6.62) | 35.67 (±9.68) | 54.58 (±7.98) | 69.66 (±4.33) | 31.08 (±3.81) | 69.33 (±3.78) |
| CNN-Trans. | ✗ | 35.75 (±3.51) | 3.06 (±4.79) | 22.35 (±2.51) | 57.93 (±5.14) | 8.37 (±4.35) | 58.63 (±4.62) |
| FFCL | ✗ | 39.19 (±6.88) | 6.28 (±8.88) | 21.20 (±7.86) | 70.69 (±2.46) | 41.36 (±3.12) | 72.54 (±2.93) |
| ST-Trans. | ✗ | 40.00 (±3.29) | 8.60 (±4.49) | 37.93 (±4.59) | 61.03 (±1.87) | 12.48 (±2.33) | 60.67 (±1.52) |
| BIOT | ✗ | 57.58 (±3.03) | 20.12 (±2.12) | 23.94 (±0.40) | 66.55 (±2.74) | 30.67 (±3.30) | 51.66 (±2.61) |
| LaBraM | ✗ | 76.25 (±2.31) | **64.07 (±3.04)** | 76.14 (±2.12) | 66.09 (±1.94) | 29.81 (±1.35) | 64.72 (±2.43) |
| CBraMod | ✗ | 72.54 (±0.22) | 41.35 (±0.27) | 66.45 (±0.25) | 77.50 (±0.22) | **82.32 (±0.28)** | **85.75 (±0.25)** |
| NeuroLM | ✓ | 68.45 (±3.04) | 51.94 (±4.61) | 68.39 (±2.97) | 63.45 (±2.25) | 23.19 (±1.74) | 66.57 (±1.92) |
| UniMind | ✓ | **78.95 (±1.34)** | 61.53 (±1.58) | 75.40 (±1.77) | **78.67 (±1.43)** | 57.33 (±1.12) | 78.65 (±1.67) |

Table 16: Performance comparison on SEED-IV Zheng et al. (2019) and SleepEDF Kemp et al. (2000) datasets. "MT" denotes multi-task learning usage. Underlining shows best multi-task results; **bold** shows best overall.

| Model | MT | SEED-IV | | | SleepEDF | | |
|---|---|---|---|---|---|---|---|
| | | B-Acc | Kappa | F1-W | B-Acc | Kappa | F1-W |
| SPaRCNet | ✗ | 29.88 (±1.37) | 6.45 (±0.92) | 32.05 (±1.48) | 60.16 (±1.21) | 65.31 (±1.36) | 58.61 (±1.44) |
| ContraWR | ✗ | 38.38 (±1.61) | 14.83 (±1.26) | 40.21 (±1.43) | 67.05 (±1.67) | 70.46 (±1.52) | 66.92 (±1.58) |
| CNN-Trans. | ✗ | 35.21 (±1.42) | 10.92 (±1.17) | 36.57 (±1.39) | 60.29 (±1.35) | 66.12 (±1.21) | 58.96 (±1.30) |
| FFCL | ✗ | 37.81 (±1.55) | 15.35 (±1.44) | 39.76 (±1.61) | 65.66 (±1.33) | 70.13 (±1.27) | 64.79 (±1.45) |
| ST-Trans. | ✗ | 36.93 (±1.46) | 15.72 (±1.30) | 36.95 (±1.50) | 55.17 (±1.49) | 67.32 (±1.38) | 53.18 (±1.42) |
| BIOT | ✗ | 36.19 (±1.38) | 23.21 (±1.47) | 42.76 (±1.52) | 64.95 (±0.26) | 71.32 (±0.32) | 60.91 (±0.40) |
| LaBraM | ✗ | **47.63 (±0.73)** | **28.77 (±0.60)** | **49.14 (±0.82)** | 68.96 (±0.75) | 75.49 (±0.68) | 87.30 (±0.71) |
| CBraMod | ✗ | 39.87 (±0.19) | 17.60 (±0.20) | 36.89 (±0.22) | 69.40 (±0.20) | 74.86 (±0.26) | 87.23 (±0.24) |
| NeuroLM | ✓ | 32.30 (±1.69) | 9.46 (±1.53) | 34.65 (±1.66) | 56.40 (±0.60) | 68.76 (±0.59) | 54.02 (±0.63) |
| UniMind | ✓ | 45.56 (±0.71) | 24.06 (±0.58) | 43.58 (±0.77) | **72.98 (±0.66)** | **76.95 (±0.63)** | **88.23 (±0.69)** |

Table 17: Performance comparison on SHHS Quan et al. (1997) and SHU Ma et al. (2022) datasets. "MT" denotes multi-task learning usage. Underlining shows best multi-task results; **bold** shows best overall.

| Model | MT | SHHS | | | SHU | | |
|---|---|---|---|---|---|---|---|
| | | B-Acc | Kappa | F1-W | B-Acc | Kappa | F1-W |
| SPaRCNet | ✗ | 63.93 (±0.97) | 63.47 (±0.91) | 61.82 (±0.95) | 62.15 (±0.78) | 25.32 (±0.81) | 62.15 (±0.74) |
| ContraWR | ✗ | 67.01 (±1.02) | 69.44 (±1.09) | 56.80 (±0.96) | 62.13 (±0.83) | 24.15 (±0.88) | 57.51 (±0.86) |
| CNN-Trans. | ✗ | 64.63 (±0.92) | 67.83 (±0.87) | 63.78 (±0.90) | 56.25 (±0.76) | 12.48 (±0.80) | 55.86 (±0.73) |
| FFCL | ✗ | 67.59 (±0.89) | 69.48 (±0.93) | 67.07 (±0.90) | 62.82 (±0.74) | 25.67 (±0.78) | 62.78 (±0.76) |
| ST-Trans. | ✗ | 64.63 (±0.95) | 67.18 (±0.91) | 63.30 (±0.94) | 63.39 (±0.70) | 26.76 (±0.73) | 63.25 (±0.75) |
| BIOT | ✗ | 72.22 (±0.73) | **78.35 (±0.78)** | 83.96 (±0.76) | 59.16 (±0.61) | 16.68 (±0.66) | 55.51 (±0.64) |
| LaBraM | ✗ | 71.69 (±0.68) | 77.07 (±0.72) | 82.90 (±0.69) | **67.90 (±0.64)** | 32.85 (±0.66) | 67.84 (±0.63) |
| CBraMod | ✗ | 70.93 (±0.21) | 71.45 (±0.27) | 79.36 (±0.24) | 66.52 (±0.20) | **74.93 (±0.24)** | **73.67 (±0.22)** |
| NeuroLM | ✓ | 59.15 (±1.23) | 61.68 (±1.15) | 63.54 (±1.10) | 59.36 (±0.78) | 16.91 (±0.75) | 56.03 (±0.73) |
| UniMind | ✓ | **74.00 (±0.62)** | 76.63 (±0.66) | **84.20 (±0.69)** | 65.77 (±0.60) | 31.54 (±0.63) | 65.73 (±0.59) |

to using the randomized pool. Furthermore, the model can correctly interpret semantically similar prompts that were never seen during training, confirming that it learns the underlying task semantics rather than overfitting to specific phrasing.

**Instruction Templates for the SHU Dataset**

1. This segment of EEG signal can reflect the subject's behavioral actions. Please determine the type of action based on the provided EEG signal? [Left hand, Right hand]

2. The given EEG signal is indicative of the subject's movements. Can you identify the action type from the EEG data? [Left hand, Right hand]

3. Analyze this EEG signal to discern the subject's physical actions. What is the action type shown? [Left hand, Right hand]

4. ... (similar instructions)

**Instruction Templates for the SEED Dataset**

1. Given this EEG signal, which emotion does it reflect? [positive, negative, or neutral]

2. Based on this EEG signal, please identify the emotion it represents. [positive, negative, or neutral]

3. From this EEG signal, can you determine which emotion it corresponds to? [positive, negative, or neutral]

4. ... (similar instructions)

**Instruction Templates for the SEED-IV Dataset**

1. Given this EEG signal, which emotion does it reflect? [neutral, sad, fear, happy]

2. Based on this EEG signal, please identify the emotion it represents. [neutral, sad, fear, happy]

3. From this EEG signal, can you determine which emotion it corresponds to? [neutral, sad, fear, happy]

4. ... (similar instructions)

**Instruction Templates for the TUAB Dataset**

1. This EEG signal may indicate abnormal conditions. Based on this signal, determine if there is an abnormality. Choose one: [Normal, Abnormal]

2. Analyze this EEG signal to assess whether it reflects an abnormal condition. Please select one: [Normal, Abnormal]

3. This EEG signal could suggest abnormal brain activity. Determine if the signal is normal or abnormal: [Normal, Abnormal]

4. ... (similar instructions)

---

**Instruction Templates for the TUEV Dataset**

1. This EEG signal reflects epileptic events. Please determine the epileptic state based on this signal.
2. Analyze this EEG signal to classify the epileptic state.
3. This EEG signal may indicate epileptic activity. Based on the signal, identify the epileptic state.
4. ... (similar instructions)

---

**Instruction Templates for the TUSL Dataset**

1. This EEG signal reflects a slow event. Based on this signal, please determine the state. Choose one: [bckg, seiz, slow]
2. Analyze this EEG signal to classify the state it indicates. Select one: [bckg, seiz, slow]
3. This EEG signal may suggest a slow event. Determine the corresponding state from the options: [bckg, seiz, slow]
4. ... (similar instructions)

---

**Instruction Templates for the SHHS , SleepEDF and HMC Dataset**

1. The EEG signal provides insights into sleep stages. Which sleep phase does it most likely correspond to? Choose one: [Sleep stage W, Sleep stage N1, Sleep stage N2, Sleep stage N3, Sleep stage R]
2. Sleep phases can be inferred from EEG signals. Given the signal, which phase is it most likely indicating? Pick one: [Sleep stage W, Sleep stage N1, Sleep stage N2, Sleep stage N3, Sleep stage R]
3. This EEG signal reflects brain activity during sleep. Which sleep stage does it most likely represent? Select one: [Sleep stage W, Sleep stage N1, Sleep stage N2, Sleep stage N3, Sleep stage R]
4. ... (similar instructions)

---

**Instruction Templates for the Workload Dataset**

1. This is an EEG signal. Is this brainwave showing high workload or low workload? [high, low]
2. Here's an EEG signal. Does it represent a high workload or a low workload on the brain? [high, low]
3. This EEG signal is given. Is the workload indicated here high or low? [high, low]
4. ... (similar instructions)

### A.8 DISCUSSION

In this work, UniMind establishes a promising approach for unified multi-task EEG decoding by leveraging a Neuro-Language Connector to align complex spatiotemporal EEG features with large language models, and a Task-aware Query Selection Module to dynamically adapt to heterogeneous decoding tasks. Our extensive experiments across ten datasets demonstrate that UniMind consistently outperforms previous multi-task models and achieves comparable or superior results to single-task approaches, highlighting the effectiveness of integrating LLMs in brain signal decoding fOr multi-task learning.

However, UniMind remains limited in some respects. At present, it has been trained exclusively on non-invasive EEG data. In the future, we aim to progressively extend the framework to additional modalities such as fMRI and MEG, in order to establish a more comprehensive and robust foundation model for multi-modal brain decoding.

