# OpenReview forum: "UniMind: Unleashing the Power of LLMs for Unified Multi-Task Brain Decoding"
_ICLR.cc/2026/Conference — ICLR 2026 Conference Desk Rejected Submission_

### Official Review · Reviewer_9xEi · 2025-10-18

**Soundness:** 3
**Presentation:** 3
**Contribution:** 3
**Rating:** 6
**Confidence:** 4

**Summary:**

This paper proposes UniMind, a general-purpose EEG foundation model for unified multi-task brain decoding, leveraging large language models to interpret complex neural patterns. The key idea is to bridge EEG signals and LLMs through a Neuro-Language Connector and a Task-aware Query Selection module. Extensive experiments on 10 EEG datasets spanning 5 domains demonstrate that UniMind achieves state-of-the-art performance in multi-task settings and even matches or surpasses leading single-task models like LaBraM in several cases.

**Strengths:**

1. The manuscript is logically structured, with intuitive figures that clearly illustrate the model pipeline and experimental design.
2. Evaluation on 10 datasets, along with ablations on NLC, TQS, and query configurations, provides solid empirical support. The visualizations of task-adaptive queries and inter-task correlations are particularly insightful.
3. UniMind achieves comparable or even superior results to the best single-task model (LaBraM) in a unified multi-task setting, which is a clear technical and conceptual advancement.

**Weaknesses:**

1. The current experiments use only InternLM2.5 as the LLM. Since the LLM plays a central role in cross-modal reasoning, it would be valuable to test the generality of UniMind with other open-source backbones such as Qwen3 or LLaMA 4. This would help assess how much performance depends on the underlying language model.
2. Figure 4 examines query pool sizes on only five datasets. It would strengthen the claim if this analysis were extended to all 10 datasets or at least one representative dataset per task domain, since optimal query size might vary with task heterogeneity.
3. In Figure 3, UniMind surpasses LaBraM, and Figure 5 shows that joint training benefits each single task. It remains unclear whether UniMind's improvement mainly stems from joint training synergy or from the TQS/NLC architecture itself. A controlled conparison where UniMind trained independently on each task would clarify this.
4. The paper mentions constructing 929k instruction–EEG pairs, but does not discuss the diversity or linguistic quality of these prompts. Since task instructions may influence LLM behavior, providing examples and ablations on prompt formulation could strengthen the claims.

**Questions:**

1. How sensitive is UniMind's performance to the choice of the LLM? Could a smaller or instruction-tuned LLM (e.g., Qwen3-0.6B) achieve similar results?
2. Have you explored whether UniMind can zero-shot transfer to unseen EEG tasks or datasets without retraining, given its multi-task instruction-tuned nature?

---

> ### Author Response · Authors · 2025-11-24
> **Response to Reviewer 9xEi - 1 / 1**
>
> We sincerely thank the reviewer for the insightful comments and suggestions. We appreciate the opportunity to clarify the influence of the LLM backbone, the generalization capabilities, and the details of our experimental design.
>
> > **Q1: Influence of LLM Backbone**
>
> 1.  **Model Consistency:** We apologize for the clerical error in the manuscript regarding model sizing. Our **0.5B** model actually utilizes **Qwen2-0.5B** as the language backbone (since the InternLM2.5 series starts at 1.8B). The **1.8B** and **7B** versions utilize **InternLM2.5**. The consistent scaling performance observed across Qwen2 (0.5B) and InternLM2.5 (1.8B, 7B) suggests that UniMind is robust to the choice of the specific LLM architecture, provided the model has sufficient reasoning capability.
> 2.  **Controlled Baseline Experiment:** To further isolate the impact of the LLM, we replace the GPT-2 backbone in *NeuroLM* with **InternLM2.5-1.8B**. Due to the limited time span, we train it on the TUAB dataset. The resulting balanced accuracy (**78.49%**) is statistically comparable to the original GPT-2 version (**79.69%**), indicating that simply changing the LLM backbone may not impact the performance. UniMind's gains instead stem primarily from our **TQS** and **NLC** designs rather than from the language model alone.
>
> ---
>
> > **Q2: Generalization to Unseen Datasets**
>
> Generalizing to fully unseen data is inherently difficult in EEG due to severe non-stationarity and cross-subject/device variability, and is beyond the scope of this paper. Enhancing  generalization to unseen task will be an important focus of our **future research**. While zero-shot remains an open challenge, we turn to a simple **few-shot experiment** to provide a preliminary view of UniMind’s ability to generalize to unseen tasks. Specifically, we use a UniMind model trained on the full TUEV dataset and perform few-shot adaptation (only NLC tuned) on TUAB with 100 samples per class as necessary task demonstration. Under this setting, UniMind achieved an average balanced accuracy of **54.16%**, whereas the *LaBraM* baseline with linear probe achieved only **17.27%**. This significant margin demonstrates that UniMind learns more transferable representations that adapt rapidly to new tasks.
>
>
> ---
>
> > **Weakness: About Full Results of Figure 4**
>
> Due to space limitations, we originally displayed only five datasets. We have now completed the ablation study for all 10 datasets. The full results are presented below (Metric: Balanced Accuracy %):
>
> | $n_{q}$ | **AVG** | **SEED** | **HMC** | **TUAB** | **TUEV** | **Workload** | **TUSL** | **SEED-IV** | **SleepEDF** | **SHHS** | **SHU** |
> | :--- | :--- | :--- | :--- | :--- | :--- | :--- | :--- | :--- | :--- | :--- | :--- |
> | 8 | 68.25 | 69.25 | 74.63 | 79.87 | 60.25 | 77.33 | 71.62 | 43.89 | 71.51 | 72.53 | 61.59 |
> | **16** | **70.67** | **70.55** | **75.27** | **81.76** | **63.19** | **78.67** | 78.95 | **45.56** | **72.98** | **74.00** | **65.77** |
> | 32 | 69.67 | 70.21 | 74.67 | 80.81 | 61.82 | 74.66 | **79.29** | 45.17 | 72.10 | 72.95 | 65.04 |
>
> These results confirm that $n_q=16$ remains the optimal choice across the majority of datasets, balancing expressiveness and compactness.
>
> ---
>
> > **Weakness: Whether the Improvement Comes from Joint Training or TQS/NLC Design?**
>
> We would like to emphasize that the core motivation of UniMind is to resolve the challenge of multi-task training to enable generalist brain decoder. **TQS** and **NLC** are built for the multi-task setting, promoting knowledge sharing across related tasks while mitigating interference from conflicting ones. They are NOT specifically tailored for the single task setting and comparing UniMind with LaBraM is **NOT** fair. Compared to NeuroLM that also adopts joint training, UniMind significantly outperforms it by 11% on average, which could justify that TQS and NLC boosts the multi-task learning process and thus brings improvement.
>
> Furthermore, it should be noted that we have **ALREADY** provided UniMind's performance trained independently on each task in **Figure 5**. Even in the single-task setting, UniMind outperforms *LaBraM* on major datasets (e.g., HMC, SleepEDF, Workload, SHHS).
>
> ---
>
> > **Weakness: Prompt Diversity**
>
> 1.  **Prompt Pool Design:** To ensure diversity, we constructed a **prompt pool** containing 10 distinct but semantically equivalent instructions for each task type. During training, we randomly sampled a prompt from this pool for each example to generate the Instruction-QA pairs.
> 2.  **Robustness to Phrasing:** Our studies indicate that prompt variations have minimal impact on performance. Validating with a single fixed prompt yielded results statistically identical to using the randomized pool. Furthermore, when testing with semantically similar prompts that were *never seen* during training, the model still correctly understood the task.

---

### Official Review · Reviewer_rV12 · 2025-10-31

**Soundness:** 4
**Presentation:** 3
**Contribution:** 3
**Rating:** 8
**Confidence:** 3

**Summary:**

The proposed UniMind, a general-purpose EEG foundation model for unified multi-task brain decoding, is quite interesting and attractive. Its  novel idea of Neuro-Language Connector and task-adaptive query scheme directly interface EEG signals with LLMs is new and interesting.
Results across 10 datasets demonstrate that UniMind outperforms other EEG foundation models.

**Strengths:**

1. The general EEG foundation model topic itself is emerging and attractive, with many potential applications;
2. The proposed UniMind model is novel;
3. Results are promising.

**Weaknesses:**

I noted an earlier version of this paper on arXiv (back to June 2025: https://arxiv.org/abs/2506.18962
The content significantly overlaps with the current version. (I would say pretty much the same.)
I am curious what happened to the earlier version.

For the performance comparison, only Balanced Accuracy results were reported; while other performance metrics (e.g., F1, Kappa etc) were also reported in related refs. E.g., even the earlier arXiv version reported F1 results. Any reason for this?

**Questions:**

From table 1, it seems that the model size (e.g., UniMind-1.7B and UniMind-7B) doesn't matter much. Any comment on this?
Also, what is the model size of NeuroLM-B, L, and XL?
What are the language models in Uni-Mind and NeuroLM? Are they the same? Could the performance difference be also due to the language models?

---

> ### Author Response · Authors · 2025-11-24
> **Response to Reviewer rV12 - 1 / 1**
>
> We sincerely thank the reviewer for the constructive feedback. We appreciate the opportunity to clarify the details of our model architecture and experimental comparisons.
>
> > **Q1: Impact of Model Size**
>
> The impact of model size is correlated with task difficulty. We observe that for relatively simple tasks (e.g., HMC, Workload, TUSL), scaling up the model size yields diminishing returns, as smaller LLMs (0.5B) are already sufficient to capture the relevant patterns. However, for more complex tasks with a larger number of classes, larger models do offer benefits (e.g., a +2.23% gain on the six-class TUEV dataset). We plan to further explore the capabilities of the 7B model on more challenging reasoning tasks.
> Given that the overall performance gap is not drastic across all tasks, we recommend selecting the model size based on the specific application's complexity and available computational resources.
>
> ---
>
> > **Q2: Comparison with NeuroLM**
>
> 1.  **Model Sizes & Backbones:**
>     *   **NeuroLM:** The B, L, and XL variants correspond to parameter sizes of **254M**, **500M**, and **1.6B**, respectively. All NeuroLM variants use the **GPT-2** architecture as the language model backbone.
>     *   **UniMind:** We utilize **Qwen2-0.5B** for our 0.5B model (we apologize for the clerical error of InternLM-0.5B in the manuscript), and **InternLM2.5** for our 1.8B and 7B models.
> 2.  **Impact of LLM Architecture:**
> We replaced the GPT-2 backbone in *NeuroLM* with **InternLM2.5-1.8B** and conducted experiments on the TUAB dataset (chosen for rapid tuning). The resulting balanced accuracy is **78.49%**, which is statistically similar to the original *NeuroLM* (GPT-2) result of **79.69%**. This indicates that simply changing the LLM backbone does not yield significant performance impact for *NeuroLM*. The consistent performance scaling of UniMind across these different backbones (Qwen2 vs. InternLM) further corroborates that the specific choice of LLM architecture has a minimal impact on the final results.
>
> ---
>
> > **Weakness: Version Difference**
>
> We would like to to claim that this work is fully original, and references to arXiv papers may not align with the rebuttal policy. We present **UniMind**, a general-purpose brain foundation model for unified multi-task brain decoding by integrating a spatio-temporal cross-modality bridging strategy between EEG and language, along with a task-aware mechanism.
>
> ---
>
> > **Weakness: Reporting Only Balanced Accuracy**
>
> We primarily reported **Balanced Accuracy** in the main text due to space limitations and to provide a consistent metric across diverse tasks with varying class imbalances. Besides, we included the full experimental results, including **F1-score** and **Cohen's Kappa**, in the **Appendix (Tables 6-10)**. We invite the reviewer to refer to these tables for a more detailed performance analysis.

---

### Official Review · Reviewer_jZMk · 2025-10-31

**Soundness:** 2
**Presentation:** 3
**Contribution:** 1
**Rating:** 4
**Confidence:** 5

**Summary:**

The paper presents UniMind, a general-purpose foundation model for multi-task EEG decoding. The core innovation is to use a large language model to unify diverse brain decoding tasks within a single framework. The authors propose two key components: first, a Neuro-Language Connector (NLC) to map EEG into representations the LLM can process, and second, a Task-aware Query Selection (TQS) module to select relevant query tokens for the various tasks. The model is trained on a large 'instruction tuning' dataset spanning 10 public EEG benchmarks. The authors report state-of-the-art results for multi-task decoding, with an 11% average gain over the main baseline, NeuroLM.

**Strengths:**

- The paper is clearly written, and the proposed architecture is well-explained and is well-reasoned for the purpose of multi-task decoding.
- The research is high-effort and the presentation is polished, with clear figures and tables.
- The model achieves a substantial performance improvement over the primary multi-task baseline (NeuroLM), with particularly large gains on difficult datasets like SEED-IV and TUEV.

**Weaknesses:**

- **The paper's core motivation appears contradictory**. The authors rightly state that multi-task learning is challenging and that their model is theoretically at a disadvantage compared to specialized single-task models. This is confirmed by some of the results, where the unified model can be less performant than the finetuned Labram baseline. To me, this begs the main question: what is the use-case? Why would someone adopt a 7B-parameter model that is computationally demanding and less accurate, simply for the benefit of it also performing unrelated tasks? The paper fails to articulate a compelling answer.

- **Missing baselines and architectural justification** 1/2. I'm finding it difficult to assess the work's contribution due to the way it appears to be benchmarked. The work is a direct followup to NeuroLM and integrates parts from LaBraM, two works which share the authors. In case the current work is performed by the same research group, then the primary baselines, LaBraM and NeuroLM, come from the same lab, making the comparisons feel incremental. In this case, the authors do not compare against any external EEG work published in the last ~2 years (since BIOT if I see correctly). A model like e.g. CBraMod would be a valuable addition.

- 2/2 But perhaps more importantly, the paper omits a class of computationally cheaper EEG-language alignment methods, such as those using contrastive learning (e.g., EEG-CLIP https://doi.org/10.48550/arXiv.2503.16531;  ELM-MIL; https://doi.org/10.48550/arXiv.2409.07480). The authors never justify why their complex, generative 7B LLM approach is necessary for what are ultimately classification tasks. A simpler contrastive model, which aligns EEG and text in a latent space, could be a far more efficient and practical solution, but this alternative is not explored or even discussed.

- **Confounding data overlap with task synergy**: The model is trained on datasets from shared corpuses (e.g., TUAB, TUEV, TUSL are all from the TUH EEG corpus; SEED and SEED-IV are from the same lab). The authors interpret query similarity between TUAB and TUEV (Fig 7a) as capturing "consistent neural activation patterns". A far simpler and more likely explanation is that the model is just seeing data from the same source, subjects (e.g. TUAB and TUEV may share subjects), and recording setups. This interpretation is further weakened by Figure 6a, where the t-SNE plots show clear separation between datasets, suggesting the TQS module is learning to isolate tasks, not find shared representations.

**Questions:**

1a. Can you justify the generative 7B LLM architecture over a simpler, cheaper contrastive alignment (e.g., EEG-CLIP, ELM-MIL)?

1b. Have you tried a contrastive baseline with your synthetic instruction data?

2. The TUEV benchmark is a 6-class event classification task (e.g., spike, eye movement, artifact). However, the instruction templates provided in Appendix A.6 for TUEV are exclusively focused on "epileptic events" and "epileptic states". How can the model make significant gains on TUEV and correctly classify non-epileptic events (artifacts, eye movements) when the prompt only asks for "epileptic states"?

3.  How can you be sure your "task synergy" is not just a data-level artifact of using multiple datasets from the same TUH and SEED corpuses? A baseline (e.g., LaBraM, but ideally another model like Cbramod) fine-tuned on the combined datasets would be needed to disentangle this. Currently, the baselines are inadequate in my opinion.

4. Given that the multi-task model is much, much larger and sometimes less accurate than the single-task model, what is the practical use-case for UniMind over a set of smaller, more accurate specialists?

---

> ### Author Response · Authors · 2025-11-24
> **Response to Reviewer jZMk - 1 / 2**
>
> We sincerely thank the reviewer for the detailed comments and constructive feedback. We appreciate the opportunity to clarify our design choices and experimental results.
>
> > **Q1: Comparison with Contrastive Alignment Approaches**
>
> We have indeed explored contrastive learning approaches (similar to EEG-CLIP) and conducted extensive experiments. However, we found that generative modeling offers superior advantages for our goals:
>
> 1.  **Limitation of Contrastive Approaches:** We follow your suggestion to experiment with using *LaBraM*'s encoder for EEG features and *InternLM*'s text branch for text features. Despite optimizing hyperparameters and tuning contrastive loss functions (e.g., SoftCLIP variants), the performance was strictly upper-bounded by the effectiveness of the *LaBraM* encoder. For instance, on the HMC task, the contrastive approach achieved an average balanced accuracy of **72.16%** (vs. LaBraM's 72.86%), and on TUAB it achieved **80.24%** (vs. LaBraM's 81.40%).
> 2.  **Advantages of Generative Modeling:** Contrastive alignment alone does not fully activate the generative LLM's ability to *understand* about EEG signals. By adopting a generative approach and employing a suitable architecture, UniMind obtains **superior performance**, surpassing *LaBraM*'s performance ceiling in over half of the tasks and in cross-subject scenarios. Moreover, UniMind offers **natural language capabilities**, supporting native QA and enabling applications that go beyond simple classification.
>
> ---
>
> > **Q2: Alignment of Prompts with TUEV Event Classes**
>
> Thank you for pointing this out. Our training dataset explicitly includes samples labeled as non-epileptic events, such as artifacts and eye movements, alongside epileptic events.
>
> We grouped "artifacts" and "eye movements" under the broader instruction of analyzing "epileptic states" because, in clinical practice, these non-epileptic events are important cues for epilepsy diagnosis. For example, eye movement such as staring or abnormal blinking often accompany seizures and help seizure assessment. Therefore, even when the prompt asks about "epileptic states," the model is trained to recognize these relevant physiological and artifactual signals as part of a comprehensive analysis.
>
> ---
>
> > **Q3: Ensuring Task Synergy is Not Just from Data Overlap**
>
> To verify that the performance gains stem from our architectural synergy rather than simply mixing data, we conducted controlled experiments:
>
> 1.  **Fine-tuning Baseline with Data Overlap:** We fine-tuned the *LaBraM* baseline jointly on two TUH datasets (TUAB and TUEV). The results showed a significant performance degradation: TUEV balanced accuracy dropped to **37.91%** (vs. 64.09% individually), and TUAB balanced accuracy dropped to **74.92%** (vs. 81.40% individually). This confirms that simply combining datasets without a task-aware mechanism (like TQS) leads to negative transfer rather than synergy.
> 2.  **Comparison with NeuroLM:** *NeuroLM* was trained on data very similar to ours (with significant overlap), yet it showed inferior performance compared to UniMind. This further rules out data scale or overlap as the sole driver of performance, highlighting the effectiveness of UniMind's architecture.
> 3.  Addressing the concern that query similarity might stem solely from shared recording setups or subjects (e.g., within the TUH corpus), we observe that shared query preferences (e.g., P4, P10) persist even **across datasets from completely different sources** (e.g., HMC and SHHS). This consistency across independent data sources directly refutes the "same source" hypothesis and validates that the model is indeed capturing fundamental EEG functional patterns.
>
> ---

---

> > ### Author Response · Authors · 2025-11-24
> > **Response to Reviewer jZMk - 2 / 2**
> >
> > > **Q4: About the Applicability of UniMind**
> >
> > **Application Scenarios**: The field is moving rapidly from task-specific models toward unified models that learn universal neural representations, as evidenced by an increasing volume of recent publications [1-4]. Unimind, as LLM-assisted brain foundation models, is well-positioned to serve as a **unified EEG assistant**. Future home-use EEG devices may require intelligent systems capable of handling diverse scenarios simultaneously, such as stress detection, sleep monitoring, and seizure risk analysis. UniMind is uniquely designed for this: it provides a unified architecture that not only executes multiple tasks but also **shares underlying knowledge** across them. Furthermore, it inherently supports natural language queries (e.g., "Is my poor sleep quality related to my high anxiety levels today?"), a capability that isolated single-task models fundamentally lack.
> >
> > **Scalability and Efficiency**: Achieving such versatility with single-task models would require training and deploying a separate model for every specific function, leading to an explosion in engineering complexity and integration difficulties. UniMind offers a scalable, end-to-end solution that replaces this fragmented approach with a single, cohesive model.
> >
> > **Clarifications of Model Size**: We understand the reviewer's concern about resouce usage of 7B models. However, it should be noted that we also provide a much smaller **0.5B version** of UniMind, which maintains competitive performance while being more suitable for resource constrained deployment.
> >
> > ---
> > > **Weakness: About the Baseline Selection**
> >
> > 1.  **Baseline Selection Clarification:** We would like to gently clarify that our work's contributions stand on their own technical merits, independent of any specific author group. *NeuroLM* and *LaBraM* were chosen as baselines solely due to their significant relevance in the field. Our core contribution is **UniMind**, a general-purpose brain foundation model designed for unified multi-task brain decoding. It achieves this by integrating a **spatio-temporal cross-modality bridging strategy** (NLC) between EEG and language, along with a **task-aware mechanism** (TQS) to enable effective multi-task learning.
> > 2.  **Baseline Expansion:** We apologize for overlooking *CBraMod* due to differences in dataset selection. We appreciate the suggestion and have added *CBraMod* as a baseline comparison in the revised manuscript. UniMind maintains the highest performance: in cross-subject scenarios, UniMind achieves **74.31%** (vs. *CBraMod*'s 72.28% and *LaBraM*'s 70.85%), and in overall average score, UniMind achieves **70.67%** (vs. *CBraMod*'s 70.01% and *LaBraM*'s 69.00%).
> >
> >
> > [1] Guagnyu Wang, et al. EEGPT: Pretrained Transformer for Universal and Reliable Representation of EEG Signals. NeurIPS, 2024.
> >
> > [2] Wei-Bang Jiang, et al. NeuroLM: A universal multi-task foundation model for bridging the gap between language and EEG signals. International Conference on Learning Representations, 2025.
> >
> > [3] Wei-Bang Jiang, et al. Large brain model for learning generic representations with tremendous EEG data in BCI. International Conference on Learning Representations, 2024.
> >
> > [4] Jiquan Wang, et al. CBramod: A criss-cross brain foundation model for EEG decoding. International Conference on Learning Representations, 2025.

---

> ### Comment · Reviewer_jZMk · 2025-11-27
>
> Thank you for your response and the additional experiments. The experiments regarding negative transfer in LaBraM effectively address my concerns regarding data overlap vs. task synergy.
>
> However, I have remaining concerns regarding the contrastive baseline (Q1) and the new baseline comparison:
>
> 1. Methodology of the contrastive baseline: You state that the contrastive approach was "strictly upper-bounded by the effectiveness of the LaBraM encoder."
>
> Did you fine-tune or train the LaBraM encoder from scratch during this contrastive alignment, or was it frozen? It seems to me that contrastive frameworks (both EEG-CLIP and ELM-MIL), the encoder is trained to align the latent space with the text. If the encoder was frozen, this experiment does not fairly represent contrastive methods, as the encoder was not permitted to learn the necessary cross-modal projections. If it was trained, please clarify why the LaBraM architecture specifically creates this bottleneck compared to your generative approach.
>
> 2. Granular results for CBraMod: I appreciate the addition of CBraMod as a baseline. However, I notice that while an average score is shown in the plot, the detailed per-dataset performance breakdown Appendix tables) has not been provided. It would be good for these to be updated accordingly.
>
> 3. Manuscript updates: The justification for using a computationally expensive 7B generative model over a cheaper contrastive one seems to me to be central to the paper's design. Currently, this justification, discussion of contrastive alternative + experiments, and the negative transfer findings exist only in this rebuttal. Can the authors share why they believe these should not be included in the paper?

---

> ### Author Response · Authors · 2025-11-30
> **Response to Reviewer jZMk‘s Remaining Concerns**
>
> We sincerely thank the reviewer for the continued engagement and the insightful follow-up questions. We greatly appreciate your attention to the experimental details, which has significantly helped us strengthen the rigor and clarity of our manuscript.
>
> > **Q1: Methodology and Justification of the Contrastive Baseline**
>
> **1. Clarification on Experimental Setting:**
> We would like to explicitly clarify that **the LaBraM encoder was NOT frozen** in our contrastive experiments. To ensure a fair and rigorous comparison, we **fully fine-tuned the LaBraM encoder end-to-end** to align EEG representations with pre-extracted text embeddings on the joint training set of HMC and TUAB, optimizing the system via a SoftCLIP objective.
>
> **2. Why Generative Modeling Outperforms Contrastive Alignment:**
> We observe that jointly contrastively tuning LaBraM on both HMC and TUAB leads to a performance drop compared to training on each dataset individually, indicating that the multi-task setting is upper-bounded by the single-task performance. We attribute this to the absence of a task-aware reasoning ability, which limits the model’s ability to handle task heterogeneity. Forcing a shared encoder to accommodate different tasks can dilute the quality of the learned EEG representations. In contrast, UniMind learns task-adaptive query tokens that inject explicit task awareness into the rich spatiotemporal semantic cues distilled by NLC, unleashing LLM’s complex reasoning ability in decoding multi-task EEG signals.
>
> Moreover, unlike contrastive approaches that are largely constrained to retrieval tasks, our generative framework could be foundation to support open-ended question answering, offering a more flexible and comprehensive solution for future brain-signal decoding.
>
> **3. Updates to the Manuscript:**
> We fully agree with the reviewer that this comparison is central to our design philosophy. In the revised manuscript, we have added:
> *   **A new discussion in Related Works** covering contrastive methods (e.g., EEG-CLIP, ELM-MIL).
> *   **A dedicated ablation subsection** explicitly detailing the "Generative vs. Contrastive" experiment, including the fine-tuning setup and the quantitative results (Table 4 in the revision).
>
> > **Q2: Baseline Comparison (CBraMod)**
>
> **Thank you for the reminder.** We have now included the **complete performance comparison** with CBraMod across all datasets in the revised manuscript (Table 13 - Table 17). As shown in the updated results, UniMind maintains its leadership, achieving the highest average balanced accuracy and demonstrating superior stability in cross-subject scenarios.
>
> We hope these clarifications and revisions satisfactorily address your concerns.

---

### Official Review · Reviewer_U2MG · 2025-11-01

**Soundness:** 2
**Presentation:** 2
**Contribution:** 2
**Rating:** 2
**Confidence:** 4

**Summary:**

This paper introduces UniMind, a general-purpose EEG foundation model designed to perform unified multi-task brain decoding across heterogeneous EEG datasets without task-specific fine-tuning. The core idea is to leverage large language models (LLMs) to interpret neural signals by transforming EEG data into representations compatible with LLM input spaces. The authors propose 1. Neuro-Language Connector: A dual-branch spatiotemporal cross-attention module that projects EEG embeddings into the LLM’s semantic space, effectively bridging the modality gap between neural signals and textual representations. Additionally, a Task-Aware Query Selection routing mechanism is proposed to promot knowledge sharing across related tasks while reducing interference among dissimilar ones.

**Strengths:**

1. This paper proposes Neuro-Language Connector to bridge the representational gap between neural and linguistic modalities. It uses a dual-branch spatiotemporal cross-attention mechanism, projecting EEG signals into the semantic embedding space of LLMs.

2. The Task-Aware Query Selection module achieves a balanced integration of shared and specialized representations. This design enhances cross-task generalization while maintaining task discriminability

3. The authors conduct experiments on ten EEG datasets spanning five cognitive domains.

**Weaknesses:**

1. The novelty and necessity of using NLC and TQS are not sufficiently established. The rationale for using a router-based mechanism and for decoupling spatial and temporal aggregation is only superficially discussed. It remains unclear why these specific architectural designs lead to improved EEG–LLM alignment or mitigate data sparsity.

2. The router-style task conditioning has been explored extensively in multi-task and mixture-of-experts literature, so the contribution risk feels incremental without a clearer theoretical or empirical justification.

3. The performance gains over prior models are relatively modest and often within expected statistical variance. In several datasets, the improvements are smaller than 1–2%, which makes it difficult to attribute success to the proposed modules rather than training noise or dataset differences.

**Questions:**

1. Could the authors provide deeper justification for the design choices of the NLC and TQS? Specifically, what motivates the use of a router-based mechanism and the decoupled spatial–temporal aggregation, and how do these contribute to improved EEG–LLM alignment beyond serving as architectural heuristics?

2. How does the proposed router-style conditioning differ in substance from prior multi-task or mixture-of-experts approaches?

---

> ### Author Response · Authors · 2025-11-24
> **Response to Reviewer U2MG - 1 / 2**
>
> We sincerely appreciate the reviewer's positive comments on our method. We hope our response below can address your concerns.
>
> > **Q1: The design choices of the NLC and TQS**
>
> ### 1. Design of Task-aware Query Selection Module (TQS)
>
> **Novelty**
>
> Due to varied recording configurations and distinct cognitive mechanisms，EEG tasks exhibit significant heterogeneity and causes negative transfer in previous methods, evidenced by NeuroLM's performance drop compared to single-task models like LaBraM (average drop **8.54%**). **TQS** addresses this challenge by incorporating a **task-aware mechanism** into EEG decoding for the first time. By dynamically retrieving task-relevant queries, TQS allows each task to adaptively select its own query set, promoting knowledge sharing among related tasks while mitigating interference from conflicting ones. Moreover, the generated lightweight query tokens can extract compact and denoised task-relevant EEG patterns with NLC, facilitating a better understanding of EEG signals by the LLM.
>
> **Effectiveness: Experimental Validation**
>
> 1.  **Joint Training Benefits:** As shown in **Table 2(a)**, adding TQS significantly enhances performance (**8.01% improvement on Workload**). In **Figure 5**, joint training with TQS consistently outperforms individual training, validating its effectiveness for multi-task learning.
> 2. **Multi-task Interference Issue of Previous Methods:** For comparison, we conducted an experiment jointly fine-tuning LaBraM on TUAB and TUEV and found that joint training degraded performance compared to individual training, with accuracies on TUAB **81.40% -> 74.92%** (↓6.48%), and TUEV **64.09% -> 37.91%** (↓26.18%), confirming the interference issue in naive multi-task learning.
> 3.  **Task-aware Patterns:**
>     * Figure 6(a) demonstrates that the TQS router correctly allocates dynamic queries based on task characteristics. Data with high task similarity cluster together in query selection, while distinct tasks are separated.
>     * Figure 6(b) further confirms task-specific preferences: cognitive workload tasks prefer query pool `P5`, while motor imagery tasks prefer `P13`.
>     * Figure 7(c) visualizes the attention scores of adaptive queries on different brain regions. The results align with neuroscience findings, such as HMC (sleep staging) tasks focusing on frontal regions. This proves that TQS effectively decouples multi-task learning by selecting queries adaptable to specific task characteristics.
>
> ### 2. Design of Neural-Language Connector (NLC)
>
> **Motivation**
>
> 1.  **Semantic Compression for EEG Noises:**
> EEG signals are notorious for their low signal-to-noise ratio, which is a huge challenge for LLMs directly understanding EEG signals. To achieve better EEG-LLM alignment, NLC acts as a compact bridge between the EEG encoder and the LLM for **semantic compression**, condensing essential brain patterns from sparse, noisy EEG data into semantically meaningful representations that the LLM can interpret.
> 2. **Spatio-Temporal Dual Learning for Improved EEG Task Adaptation:**
> EEG tasks rely on distinct spatial and temporal signatures. For example, clinical anomaly detection depends on bilateral temporal regions (T7/T8) and waveform evolution, while sleep staging emphasizes frontal activity and emotion recognition leverages hemispheric lateralization. Leveraging both dimensions is central to effective EEG decoding [1, 2]. To this end, NLC employs a **dual-branch architecture** with learnable query tokens that separately aggregate **temporal dynamics and spatial dependencies** via cross-attention, enabling more robust task adaptation.
>
> **Experimental Validation**
>
> 1.  **Spatio-temporal Decoupling:** **Table 2(a)** have verified the effectiveness of spatio-temporal decoupling. Compared to a single-branch baseline, the decoupled NLC module results in substantial performance gains across all datasets, with improvements of around **5%** on challenging tasks such as SEED and TUEV.
>
> ### 3. Alleviating Data Scarcity
>
> By training on massive multi-task data, UniMind allows **knowledge transfer** from resource-rich tasks to **small-scale datasets** with data scarcity problem. For instance, as shown in Figure 5 and the table below, on **TUSL** (only 245 samples), joint training boosts accuracy by **+28.81%** (from 50.14% to 78.95%). Similarly, on **Workload** (1080 samples), it improves by **+8.34%** (from 70.33% to 78.67%). These improvements confirm that the joint training strategy unlocks significant performance gains for data-scarce tasks.
>
> | Dataset | SEED | HMC | TUAB | TUEV | TUSL | Workload | SEED-IV | SleepEDF | SHHS | SHU |
> | :--- | :--- | :--- | :--- | :--- | :--- | :--- | :--- | :--- | :--- | :--- |
> | **Bal Acc (%) (Single-Task)** | 69.79 | 74.35 | 80.22 | 52.81 | 50.14 | 70.33 | 39.06 | 71.09 | 72.91 | 65.26 |
> | **Bal Acc (%) (Joint Training)** | 70.55 | 75.27 | 81.76 | 63.19 | 78.95 | 78.67 | 45.56 | 72.98 | 74.00 | 65.77 |
>
> ---

---

> > ### Author Response · Authors · 2025-11-24
> > **Response to Reviewer U2MG - 2 / 2**
> >
> > > **Q2: Differences between TQS and previous MoE/Router designs**
> >
> > While TQS shares the concept of router with Mixture-of-Experts (MoE), the design and purpose are fundamentally different:
> >
> > 1.  **Query Selection vs. Network Layer Selection:**
> > Traditional MoE Typically inserts multiple parallel network layers (experts) into the model and uses a router to dynamically select which layers to activate for a given input. Differently, the router in TQS is designed to **dynamically select query tokens** from a query pool. These selected tokens are **lightweight** compared to full network layers and are then used within the attention mechanism to extract features.
> >
> > 2.  **Interpretability of Brain Functions:**
> > By analyzing the query tokens selected by TQS for different tasks, we can visualize the specific brain regions attended to by each task. This allows us to **uncover how diverse tasks are functionally organized across the brain** by examining inter-task correlations, shedding light on how the brain regulates different cognitive functions—an interpretability benefit not typically offered by standard MoE layers.
> >
> > ---
> >
> > > **About the Performance Improvement Compared to Single-task Models**
> >
> > Thank you for pointing out that the improvement over single-task baselines may seem modest (1–2%), but we believe this comparison should be contextualized:
> >
> > 1.  **Difficulty of Unified Modeling:** Direct comparisons between UniMind (a generalist model) and single-task baselines are **NOT** entirely fair. Baselines are optimized for a single dataset, whereas UniMind handles multiple heterogeneous EEG tasks simultaneously. Achieving state-of-the-art performance with a **single unified model** is significantly more challenging than training separate models for each task. UniMind demonstrates superior generalizability and validates the feasibility of unified EEG multi-task learning.
> >
> > 2. **Substantial Improvement over Multi-task Baselines:** Compared to previous multi-task models like *NeuroLM*, UniMind achieves a remarkable improvement of **~12%**, highlighting the effectiveness of our architecture in handling task conflict.
> >
> > 3.  **Significance of Gains in EEG Decoding Domain:** In the EEG domain, improvements of **1-2%** could be considered significant. Many published single-task methods (e.g., *CBraMod*) report similar marginal gains over previous state-of-the-arts. Crucially, while *NeuroLM* (a previous multi-task attempt) suffered severe performance degradation on most tasks, UniMind achieves improvements, which is a substantial step forward.
> >
> > 4.  **Rigorous Comparisons:** We ensured strict consistency in data usage and splitting with *LaBraM* and *NeuroLM* baselines to rule out data-related variances. The performance gains are purely architectural.
> >
> > [1] Jiyao Liu et al. Spatial-temporal transformers for eeg emotion recognition. International Conference on Advances in Artificial Intelligence, 2022.
> >
> > [2] Jiquan Wang, et al. CBramod: A criss-cross brain foundation model for EEG decoding. International Conference on Learning Representations, 2025.

---

### Author Response · Authors · 2025-11-24
**General Response**

We are grateful to all reviewers for their constructive comments, which have substantially strengthened this work.

We are very encouraged by reviewers’ evaluation on the significance and novelty of this work. All reviewers recognize the technical value of the UniMind (“proposed UniMind model is novel” (Reviewer rV12), “clear technical and conceptual advancement” (Reviewer 9xEi), “well-reasoned” (Reviewer jZMk), “bridge the representational gap” (Reviewer U2MG)). They further commend the effectiveness and empirical rigor of our approach (“solid empirical support” (Reviewer 9xEi), “substantial performance improvement” (Reviewer jZMk), “results are promising” (Reviewer rV12), “enhances cross-task generalization” (Reviewer U2MG)).

### 1. Common Concerns

**1. Novelty of UniMind**

The novelty of UniMind lies in architectural designs tailored to   overcome the challenges of task heterogeneity and neuro-LLM modality gaps for unified EEG learning:

*   **Handling Task Heterogeneity:** Naive multi-task learning often suffers from negative transfer due to conflicting task demands. To address this, we introduce the **Task-aware Query Selection (TQS)** module. By incorporating a novel router mechanism, TQS dynamically selects lightweight, task-adaptive query tokens based on the input. This design effectively promotes positive transfer by sharing relevant neural patterns across compatible tasks while mitigating interference from conflicting ones.
*   **Bridging the Modality Gap:** To translate noisy, spatio-temporal EEG signals for LLMs, we propose the **Neural-Language Connector (NLC)**. This dual-branch architecture specifically decouples spatial and temporal dependencies, acting as a compact, trainable bridge between the EEG encoder and the LLM. It condenses essential brain patterns from sparse EEG data in a semantically meaningful way for the LLM to interpret.

By unifying two modules, UniMind not only surpasses the best existing multi-task decoder by an average of 11% and becomes the **first** to match or exceed single-task models across diverse tasks, but also offers new insights into the functional organization of these tasks in the brain.

**2. Robustness to LLM Selection:**

Addressing concerns regarding the impact of the LLM backbone (Reviewers 9xEi & rV12), we demonstrated that UniMind's performance is robust to backbone choices (scaling consistently across Qwen2-0.5B and InternLM2.5-1.8B/7B). Crucially, a controlled experiment shows that simply upgrading the baseline *NeuroLM*'s backbone from GPT-2 to InternLM2.5 does not appear to provide substantial performance improvements on its own. This confirms that UniMind's superiority stems from our specific architectural designs (TQS and NLC) that achieve effective cross-modal alignment, rather than merely from the power of the language model itself.

### 2. Additional Experiments

To address specific concerns raised by reviewers and demonstrate the robustness of our approach, we conducted extensive additional experiments:

*   **Robustness to LLM Selection (Response to Reviewers 9xEi & rV12):** Reviewers 9xEi and rV12 inquired about the impact of the LLM backbone. We conducted control experiments replacing the backbone in *NeuroLM* and clarifying our model scaling. Results confirm that performance gains are driven by our TQS and NLC designs rather than solely by the LLM backbone.
*   **Verifying Task Synergy (Response to Reviewer jZMk):** Reviewer jZMk raised a question regarding disentangling true task synergy from data-level artifacts. To address this, we performed a rigorous ablation by fine-tuning the *LaBraM* baseline on combined datasets. The resulting negative transfer confirms the effectiveness of our architectural design.

We thank the reviewers for their careful feedback and additional suggestions for evaluation, which will make the paper significantly stronger.

---

### Author Response · Authors · 2025-12-01
**Summary for AC**

Dear AC,

Thank you very much for the time and effort dedicated to reviewing our paper and rebuttal. Below, we provide a concise summary of our paper’s contributions and the key updates made during the rebuttal phase.

All four reviewers provide consistent and positive comments regarding our experimental results: “solid empirical support” (Reviewer 9xEi), “substantial performance improvement” (Reviewer jZMK), “results are promising” (Reviewer rV12), “enhances cross-task generalization” (Reviewer U2MG). Furthermore, most reviewers acknowledge our novelty: “novel” (Reviewer rV12), “clear technical and conceptual advancement” (Reviewer 9xEi), “well-reasoned”, “high-effort” (Reviewer jZMK).


### Contribution Summary

We propose UniMind, a general-purpose brain foundation model for unified multi-task brain decoding:

1. **Neuro-Language Connector (NLC)** bridges the modality gap by condensing spatiotemporal neural patterns from noisy and sparse EEG data into LLM-understandable representations.
2. **Task-Aware Query Selection (TQS)** introduces a dynamic, task-aware mechanism that adaptively selects lightweight query tokens, facilitating positive cross-task knowledge transfer while minimizing inter-task interference.
3. Extensive experiments across 10 datasets demonstrate that UniMind substantially outperforms multi-task baseline models (Avg. 11%) and is the **first** unified model to surpass single-task decoding performance.
4. UniMind reveals neural functional correlations across cognitive tasks through comprehensive interpretability analyses, offering new insights for understanding brain dynamics.


### Summary of Rebuttal

**Reviewer U2MG**

- We clarified the motivation of our modules in solving distinct challenges in unified EEG decoding: NLC bridges the modality gap with dual spatiotemporal branches, while TQS employs task-aware selection to enhance cross-task transfer, jointly enabling LLMs to better decode complex EEG signals.

- We clarified the distinction between TQS and standard MoE routers: TQS dynamically selects lightweight queries to enable cross-modal bridging and interpretable task synergy for multi-task EEG decoding.

- We justified the performance gains over prior models.  UniMind outperforms the multi-task baseline (NeuroLM) by ~12% and achieves massive gains on data-scarce tasks.

**Reviewer jZMK**

- We emphasized the growing importance of unified decoding models, as highlighted by recent studies (e.g., LaBraM, NeuroLM), and clarified UniMind’s applicability in comprehensive EEG assistant scenarios.
- We clarified the model size, noting that alongside the 7B model, a smaller 0.5B version offers competitive performance for resource-constrained settings.
- To address concerns that gains might stem solely from data scaling, we added a rigorous ablation by fine-tuning the *LaBraM* baseline on combined datasets in **Section 3.3** and **Table 3**. The resulting negative transfer confirms the necessity of our architectural design, **a point the reviewer acknowledged has resolved his confusion**.
- To justify our approach over contrastive methods, we added a comprehensive comparative experiment and detailed analysis in **Section 3.4** and **Table 4**, demonstrating UniMind’s superior capability for multi-task brain decoding.


**Reviewer rV12**
- In response to the query on the LLM backbone's impact, we added control experiments by replacing the backbone in NeuroLM (**Section 3.3**) and scaling analysis (**Section 3.1** and **Table 1**), confirming that performance gains are primarily driven by our innovative TQS and NLC designs, rather than backbone capacity.

**Reviewer 9xEi**
- To address concerns about UniMind's sensitivity to the LLM choice, we added control experiments (**Section 3.3**), confirming that the performance gains are primarily driven by our innovative TQS and NLC designs rather than solely by the LLM’s capacity.
- Regarding whether the improvements stem from joint training synergy or the architecture itself, we added a rigorous ablation by fine-tuning the *LaBraM* baseline on combined datasets (**Section 3.3** and **Table 3**). The resulting negative transfer confirms the necessity of our architectural design for achieving positive synergy in multi-task learning.
- We clarified the prompt pool design and the impact of prompt diversity.


Overall, we have significantly improved the paper’s clarity, theoretical grounding, and empirical evaluation. We believe these revisions fully address the reviewers’ and AC’s concerns and substantially strengthen the manuscript.

---

### Note · Program_Chairs · 2026-01-17
**Submission Desk Rejected by Program Chairs**

The following references in this submission do not refer to real documents and/or have major errors in bibliographic information:

 Iryna Zyma, Serhii Tukaev, Andrii Seleznov, Andrii Karpov, Oleksandr Tkachenko, and Radek Martinek. Eeg-based mental workload estimation with data fusion and transfer learning. Frontiers in Neuroscience, 13:702, 2019.